# RegBN: Batch Normalization of Multimodal Data with Regularization

**Morteza Ghahremani**[1,2]    **Christian Wachinger**[1,2]
[1]Lab for AI in Medical Imaging (AI-Med), Department of Radiology,
Technical University of Munich (TUM), Germany
[2]Munich Center for Machine Learning (MCML), Germany
{morteza.ghahremani, christian.wachinger}@tum.de

## Abstract

Recent years have witnessed a surge of interest in integrating high-dimensional data captured by multisource sensors, driven by the impressive success of neural networks in integrating multimodal data. However, the integration of heterogeneous multimodal data poses a significant challenge, as confounding effects and dependencies among such heterogeneous data sources introduce unwanted variability and bias, leading to suboptimal performance of multimodal models. Therefore, it becomes crucial to normalize the low- or high-level features extracted from data modalities before their fusion takes place. This paper introduces RegBN, a novel approach for multimodal Batch Normalization with REGularization. RegBN uses the Frobenius norm as a regularizer term to address the side effects of confounders and underlying dependencies among different data sources. The proposed method generalizes well across multiple modalities and eliminates the need for learnable parameters, simplifying training and inference. We validate the effectiveness of RegBN on eight databases from five research areas, encompassing diverse modalities such as language, audio, image, video, depth, tabular, and 3D MRI. The proposed method demonstrates broad applicability across different architectures such as multilayer perceptrons, convolutional neural networks, and vision transformers, enabling effective normalization of both low- and high-level features in multimodal neural networks. RegBN is available at https://mogvision.github.io/RegBN.

## 1 Introduction

Multimodal models, which adeptly fuse information from a diverse range of sources, have yielded promising results and found many applications such as language and vision [27, 29, 44, 17, 51], multimedia [2, 35, 12], affective computing [63, 41, 64, 48, 56], robotics [23, 49, 24], human-computer interaction [43, 41], and healthcare diagnosis [40, 60, 36, 14, 61]. Multimodal machine learning presents distinctive computational and theoretical research challenges due to the diversity of data sources involved [25, 26]. The impressive versatility and efficacy of multimodal neural network models can be attributed to their ability to effectively integrate and leverage heterogeneous data.

Multimodal models process heterogeneous information obtained from multisource sensors by extracting their features. Subsequently, the extracted features are fused at different levels (including early, middle, or late fusion [58, 35, 36]) to address specific tasks, such as classification, recognition, description, and segmentation [25, 26, 12, 1, 10]. Heterogeneous information from multisource sensors, however, is susceptible to confounding effects caused by extraneous variables or multiple distributions [30, 7, 5, 11, 57, 32, 6]. Confounding variables pertain to external factors that introduce bias (either positive or negative) in the relationship between the variables being studied [42, 50]. The complexity of confounders emerges from their potential pervasiveness across diverse data modal-

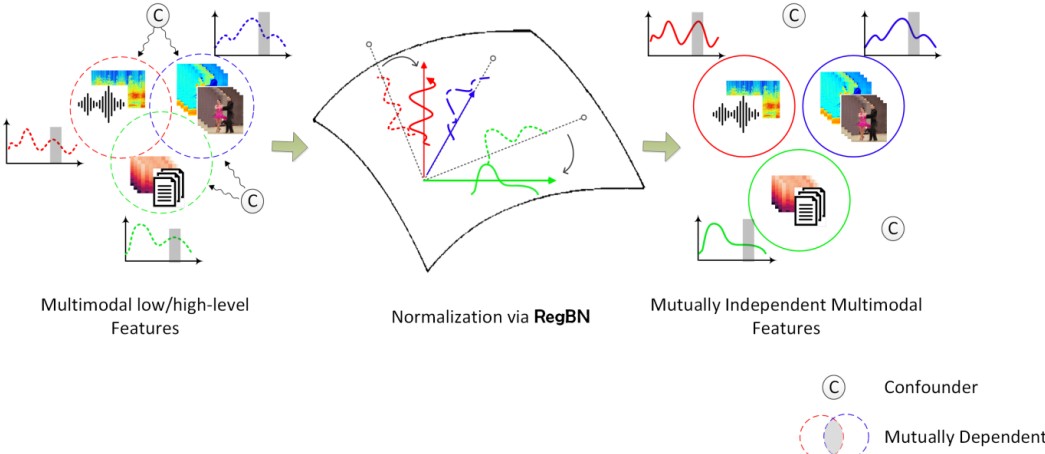

Multimodal low/high-level Features

Normalization via **RegBN**

Mutually Independent Multimodal Features

C   Confounder

Mutually Dependent

Figure 1: The presence of heterogeneous data often entails external confounding effects (denoted by 'C' in the figure) and partial dependencies, which impede the efficient training of a multimodal network. This study recommends the normalization of low- and high-level features extracted by foundation models through the use of RegBN. This allows the normalised features to be rendered independent, enabling a multimodal network to discern underlying patterns and optimize its performance.

ities. For instance, in image analysis, confounders might encompass lighting variations, while in audio classification, speaker attributes like race or gender can be confounding factors. In video parsing, backgrounds play a role, and in dementia diagnosis, the education level of patients can be a confounder. Furthermore, positive or negative correlations can exist among heterogeneous data that impact the distributions of the learned features [50, 30]. These factors pose challenges for a multimodal network to accurately uncover the actual relationship between variables for a given task. Ignoring the confounding effects and partial dependencies in features or data can result in substantial drawbacks, including the deviation of a multimedia model from its global minimum, reduction in the speed and stability of neural network training, and the generation of unreliable or misleading outcomes. Consequently, it is recommended to normalize features from different modalities before their fusion [30, 55]. In this study, we show that the normalization step can help mitigate the confounding effects and enhance the overall performance and reliability of the multimodal networks.

Normalization techniques have proven to be highly effective in enhancing the speed and stability of neural network training. By re-centering and re-scaling the input layers, the normalization methods provide a promising solution to overcome the challenges of training deep neural networks. Batch normalization (BN) [18], layer normalization (LN) [4], group normalization (GN) [62], and instance normalization (IN) [62] are popular normalization techniques that have been used in the foundation of many state-of-the-art neural networks such as ResNet [15], DenseNet [16], Inception [52], and others. The aforementioned normalization methods are designed to standardize feature distributions and do not take into account confounding effects and dependencies among features. Recent progress in multimodal learning leverages the potential of extensive multimodal data representations. CLIP [45], ALIGN [20], and MaMMUT [22] are multimodal foundations devised for images, text, audio, and video. Expanding this scope, ImageBind [13] and Gato [46] tackle diverse tasks across multiple modalities. Dealing with large-scale multimodality data, however, may introduce new obstacles such as modality heterogeneity, modality imbalance, confounding, and intermodal variabilities. These emphasize the necessity of developing a normalization technique that is dedicated to multimodal data.

To tackle these challenges, several methods [30, 55, 65] have recently been proposed that use metadata—the data that provides information about given data—for the normalization of the features in a neural network. However, these studies predominantly focus on metadata and still encounter issues related to confounding effects. Motivated by these findings, we introduce a novel normalization method for multimodal heterogeneous data, referred to as RegBN, aimed at removing confounding effects and dependencies from low- and high-level features before fusing those. Our approach entails leveraging regularization to promote independence among heterogeneous data from multisource

sensors (Figure 1). RegBN facilitates the training of deep multimodal models while ensuring the prediction of reliable results. Our key contributions are as follows:

- As a normalization module, RegBN can be integrated into multimodal models of any architecture such as multilayer perceptrons (MLPs), convolutional neural networks (CNNs), vision transformers (ViTs), and other architectures.

- RegBN possesses the capability to be applied to a vast array of heterogeneous data types, encompassing text, audio, image, video, depth, tabular, and 3D MRI.

- RegBN undergoes comprehensive evaluation on a collection of eight datasets, including multimedia, affective computing, healthcare diagnosis, and robotics.

The findings indicate that RegBN consistently leads to substantial improvements in inference accuracy and training convergence regardless of the data type or method used.

## 2    Related work

**Normalization methods in deep learning**: BN [18], LN [4], GN [62], and IN [62] are conventional normalization methods that normalize the input features by re-centering and re-scaling. Unlike such methods that process input layers separately, RegBN takes two input layers and produces corresponding mutually independent output layers. The input layers could be any $n$-dimensional low-/high-level features, metadata, or raw data.

**Confounding effect removal**: Several statistical techniques have been developed for regressing out confounders [42, 33, 38]. Such methods are basically developed for confounding effect removal in healthcare. In recent years, several studies used deep neural networks for coping with confounders [65, 30, 55]. Lu et al. [30] introduced a novel layer normalization module called 'metadata normalization' (MDN), which is designed specifically for neural networks that incorporate metadata. A closed-form solution to linear regression was developed by the authors to capture the relationship between feature layers and metadata. MDN, however, exhibits certain limitations such as its weak performance on small mini-batches and the requirement of the entire metadata for computing its matrix inverse. A penalty-based approach called PMDN [55] was proposed as an improvement over MDN. PMDN addresses the shortcomings of MDN by using learnable parameters instead of estimating MDN's matrix inverse. PMDN is designed for metadata with a small number of features due to the huge number of required learning parameters. PMDN also requires a time-consuming two-step training procedure to optimize its learnable parameters. In addition, a major limitation of PMDN is how to effectively train the learnable parameters, as these parameters are trained solely based on the model's loss function, and models are typically unaware of the presence of confounding effects.

We leverage the great potential of regularization as a solution to address the aforementioned issues. Our proposed normalization technique does not rely on learnable parameters and is capable of operating efficiently on small mini-batches. Most importantly, our approach produces more accurate results and is applicable to various types of multimodal neural networks, not limited to those incorporating metadata, and its ability to operate at both low- and high-feature levels. RegBN is the first normalization method to address the dependency and confounding issues in multimodal data, making it a unique and promising contribution to the field.

## 3    RegBN: A regularization method for normalization of multimodal data

Given a trainable multimodal neural network (e.g., MLPs, CNNs, ViTs) with multimodality backbones $\mathcal{A}$ and $\mathcal{B}$. Let $f^{(l)} \in \mathcal{R}^{b \times n}$ represent the $l$-th layer of the multimodal network for modality $\mathcal{A}$ with batch size $b$ and $n_1 \times \ldots \times n_N$ features that are flattened into a vector of size $n$. In a similar vein, we define $g^{(k)} \in \mathcal{R}^{b \times m}$ as the $k$-th layer of the multimodal network for modality $\mathcal{B}$ with $m_1 \times \ldots \times m_M$ features that are flattened into a vector of size $m$. Layers $l$ and $k$ can be positioned directly prior to the fusion step. Depending on the fusion approach employed, $f^{(l)}$ and $g^{(k)}$ can contain low-level features, high-level features, or latent representations of the multimodal model for early, middle, or late fusion, respectively (see Appendix **??**). Features $f^{(l)}$ and $g^{(k)}$ pertain to two distinct modalities and are subject to immeasurable unknown confounders. Moreover, they also share partial common information. The goal is to find potential similarities (mainly caused by confounding factors and dependencies during data collection) between these layers and then remove those (Figure 1). Mutually-independent layers can be then fused in the multimodal network for a

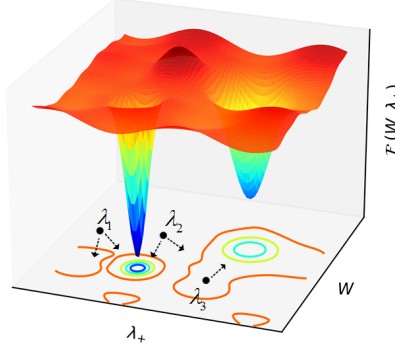
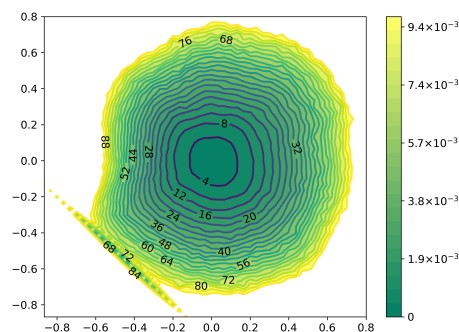

(a) Extrema in a mini-batch and seeded values $\{\lambda_1, \lambda_2, \lambda_3\}$ for finding sub-optimal $\lambda_+$

(b) 2D visualization of a multimodal LeNet model with RegBN on AV-MNIST [31]

Figure 2: (a) In a mini-batch, there is a risk of falling into local minima; we fix this problem by seeding $\lambda_+$ at different locations, detailed in Section 3.2. (b) 2D visualization of the loss surface of the multimodal SMIL model [31] on AV-MNIST (see Section 4.1). The numbers inside the plot indicate the classification loss, while the color bar reports the L1 loss of the projection weights in RegBN, i.e. $\Delta W_t$. RegBN's weights converge into their global minimum as long as the multimodal model reaches its global minimum.

given task. To this end, one can represent $f^{(l)}$ based on $g^{(k)}$ as a linear regression model

$$f^{(l)} = W^{(l,k)} g^{(k)} + f_r^{(l)}. \tag{1}$$

Here, $W^{(l,k)}$ is a projection matrix of size $n \times m$ and $f_r^{(l)}$ represents the difference between $f^{(l)}$ and its corresponding map in the domain of $g^{(k)}$, also known as the residual. Ideally, the residual map does not contain any features of $g^{(k)}$, so $f_r^{(l)}$ and $g^{(k)}$ are mutually independent. RegBN minimizes the linear relationship between these layers via

$$\hat{W}^{(l,k)} = \arg\min_{W^{(l,k)}} \left\| f^{(l)} - W^{(l,k)} g^{(k)} \right\|_2^2 \quad \textit{w.r.t.} \quad \left\| W^{(l,k)} \right\|_F = 1. \tag{2}$$

The constraint in terms of the Frobenius norm, $\left\| W^{(l,k)} \right\|_F = \sum_{i=1}^{n} \sum_{j=1}^{m} \omega_{i,j}^{(l,k)\,2} = 1$, guarantees that the network does not experience vanishing and exploding gradients during training and inference. Eq. 2 is a regularization method for ill-conditioned problems [37, 9, 34, 47]. A Lagrangian multiplier offers an equivalent formulation $\mathcal{F}$ to the optimization defined at Eq. 2:

$$\mathcal{F}(W^{(l,k)}, \lambda_+) = \left\| f^{(l)} - W^{(l,k)} g^{(k)} \right\|_2^2 + \lambda_+ \left( \left\| W^{(l,k)} \right\|_F - 1 \right). \tag{3}$$

In this equation, $\lambda_+$ is a positive Lagrangian multiplier, playing the role of a dual variable. Minimizing $\mathcal{F}(W^{(l,k)}, \lambda_+)$ over $W^{(l,k)}$ yields the projection matrix

$$\hat{W}^{(l,k)} = \left( g^{(k)\top} g^{(k)} + \hat{\lambda}_+ \mathbf{I} \right)^{-1} g^{(k)\top} f^{(l)}, \tag{4}$$

where, $\hat{\lambda}_+$ is the estimated Lagrangian multiplier obtained through minimization of the term $\left\| W^{(l,k)} \right\|_F - 1$; $\mathbf{I}$ is an identity matrix and superscript $\top$ represents the transpose operator. We employ singular value decomposition (SVD) and limited-memory Broyden-Fletcher-Goldfarb-Shanno (L-BFGS) [28] for solving Eq. 2. The solution is detailed in Appendix **??**, and the proposed method is summarized in Algorithm 1. It is worth noting that the projection weights are not directly trained by the multimodal criterion since there is a danger of being fooled by confounders. Instead, the projection weights are learned through Eqs. 2-4 and updated recursively over chunks of training data, detailed below.

### 3.1 Update of the projection matrix

The projection matrix, $W^{(l,k)}$, is calculated for every mini-batch so we recursively update that via the exponential moving average's approach [21], which decays the mean gradient and variance for each

---

**Algorithm 1** Pseudocode of RegBN

---

1: **Inputs**: $f^{(l)} \in \mathcal{R}^{b \times n}$ and $g^{(k)} \in \mathcal{R}^{b \times m}$: $l$-th and $k$-th layer of an MML, respectively

2: **Other inputs**: $t$ (timestep/mini-batch), projection weights $W^{(l,k)}$, $\Lambda_{t-1}$ (a collection of estimated $\lambda_+$ values until previous timestep), *training*;

3: Compute SVD of $g^{(k)}$: $\text{SVD}\big(g^{(k)}\big) = U\Sigma V^* = \sum_{i=1}^{m} \sigma_i u_i v_i^*$

4: Set $\mathbf{\Lambda}_t$ through Eq. 8 and given $\Lambda_{t-1}$;

5: **for** $\lambda$ in $\mathbf{\Lambda}_t$ **do**

6:     Initialize $\lambda_+$ with $\lambda$ and then apply the L-BFGS algorithm to approximate Eq. 7;

7:     Store the estimated $\lambda_+$ and the approximation error;

8: **end for**

9: Choose $\lambda_+$ with the lowest approximation error as the sub-optimal $\hat{\lambda}_+$;

10: Update $\Lambda_t$: $\Lambda_t \leftarrow \Lambda_{t-1} \cup \hat{\lambda}_t$;

11: Insert $\hat{\lambda}_+$ to Eq. 4 for computation of the projection matrix: $\hat{W}^{(l,k)} = \sum_{i=1}^{m} \big(\frac{\sigma_i u_i f^{(l)}}{\sigma_i^2 + \hat{\lambda}_+}\big) v_i$;

12: **if** *training* **then**

13:     Normalize the input feature layer using $\hat{W}^{(l,k)}$: $f_r^{(l)} \leftarrow f^{(l)} - \hat{W}^{(l,k)} g^{(k)}$;

14: **end if**

15: Update the projection weights $W_t^{(l,k)}$ via Eqs. 5 and 6;

16: **if** *not training* **then**

17:     Normalize the input feature layer using $W_t^{(l,k)}$: $f_r^{(l)} \leftarrow f^{(l)} - W_t^{(l,k)} g^{(k)}$;

18: **end if**

19: **return** $f_r^{(l)}$, $W_t^{(l,k)}$, and $\Lambda_t$

---

variable exponentially. Let $W_{t-1}^{(l,k)}$ denote the updated projection matrix until the previous timestep. We define $\Delta W_t$ as the mean absolute error between the currently predicted projection matrix and the previously updated projection matrix: $\Delta W_t = \left\| \hat{W}^{(l,k)} - W_{t-1}^{(l,k)} \right\|_1$. The projection matrix is then updated at timestep $t$ via

$$W_t^{(l,k)} = \left(1 - \gamma_t \frac{m_t}{\sqrt{\nu_t + \epsilon}}\right) \hat{W}^{(l,k)} + \gamma_t \frac{m_t}{\sqrt{\nu_t + \epsilon}} W_{t-1}^{(l,k)}. \tag{5}$$

Here, $\gamma_t$ is the learning rate of the multimodal model at timestep $t$; $m_t$ and $\nu_t$ are the first and second moments, respectively, derived from

$$m_t = \frac{\beta_1 m_{t-1} + (1 - \beta_1)\Delta W_t}{1 - \beta_1^t}, \ \nu_t = \frac{\beta_2 \nu_{t-1} + (1 - \beta_2)\Delta W_t^2}{1 - \beta_2^t}, \tag{6}$$

where $\beta_1, \beta_2 \in (0, 1)$ represent constant exponential decay rates. As mentioned in Algorithm 1, the projection matrix is updated during training only (Algorithm 1: Steps 11-13), and we use the latest updated projection weights for validation or inference (Algorithm 1: Steps 15-17).

## 3.2 Avoiding falling into local minima

Multimodal networks often contain several backbones (e.g., MLPs, CNNs, ViTs), depending on the number of multiple heterogeneous data sources. Identifying the global minima of such networks is a challenging problem, as the networks may fall into local minima. On the other hand, optimization of Eq. 3 relies on $\lambda_+$, which may introduce several local minima. Conventional regression methods treat $\lambda_+$ as a constant/hyperparameter, while predetermined hyperparameters usually increase the risk of falling into local minima. To prevent this, we adopt a mini-batch-wise approach for the estimation of $\lambda_+$. The objective is to estimate suboptimal $\lambda_+$ with the L-BFGS optimization algorithm in a way that meets

$$\hat{\lambda}_+ = \arg\min_{\lambda_+} \frac{\partial \mathcal{F}(W^{(l,k)}, \lambda_+)}{\partial \lambda_+} = \arg\min_{\lambda_+} \left( \left\| \big(g^{(k)\top} g^{(k)} + \lambda_+ \mathbf{I}\big)^{-1} g^{(k)\top} f^{(l)} \right\|_F - 1 \right). \tag{7}$$

Quasi-Newton methods like L-BFGS rely on the initial value of $\lambda_+$ for minimization of the optimization problems like Eq. 7 (see Figure 2a). Due to the limited number of local minima that exist per mini-batch, we adopt a method whereby we initialize $\lambda_+$ with multiple random values within a

Table 1: The experimental section covers a diverse range of research areas, dataset sizes, input modalities (in the form of $i$: image, $l$: language, $v$: video, $a$: audio, $s$: 3D snippet-level features, $m$: 3D MRI, $d$: depth, $t$: tabular, $f$: force sensor, $p$: proprioception sensor), and prediction tasks.

| Area | Dataset | Modalities | Samples (#) | | Prediction task |
|------|---------|-----------|-------------|------|-----------------|
| | | | Training | Test | |
| Affective computing | CMU-MOSEI | $\{v, a, l\}$ | 18,118 | 4,659 | sentiment |
| Affective computing | CMU-MOSI | $\{v, a, l\}$ | 1,513 | 686 | sentiment |
| Affective computing | IEMOCAP | $\{v, a, l\}$ | 3,515 | 938 | emotion |
| Multimedia | MM-IMDb | $\{i, l\}$ | 18,160 | 7,799 | movie genre |
| Multimedia | LLP | $\{a, v, s\}$ | 10,649 | 1,200 | video parsing |
| Multimedia | AV-MNIST (small) | $\{i, a\}$ | 1,045 | 450 | classification |
| Multimedia | AV-MNIST | $\{i, a\}$ | 60,000 | 10,000 | classification |
| Healthcare diagnosis | ADNI | $\{m, t\}$ | 1,073 | 268 | dementia diagnosis |
| Robotics | Vision&Touch | $\{i, d, f, p\}$ | 40,546 | 22,800 | contact |
| - | Synthetic dataset | $\{i, t\}$ | 10,000 | 1,000 | classification |

predetermined range $\Lambda_p = [\lambda_{p_1}, \ldots, \lambda_{p_C}]$ as well as with the median of the suboptimal $\lambda_+$ values predicted in the previous mini-batches, $\mu_{\frac{1}{2}}(\Lambda_{t-1})$:

$$\boldsymbol{\Lambda}_t = \Lambda_p \cup \mu_{\frac{1}{2}}(\Lambda_{t-1}) = [\lambda_{p_1}, \ldots, \lambda_{p_C}] \cup \mu_{\frac{1}{2}}\{[\hat{\lambda}_1, \ldots, \hat{\lambda}_{t-1}]\}. \tag{8}$$

By creating a loop over $\boldsymbol{\Lambda}_t$, we first initialize $\lambda_+$ and then estimate its true value using the L-BFGS method. We also store the approximation error and after the termination of the loop, the suboptimal value of $\lambda_+$ for a given mini-batch is determined by selecting the estimated value that yields the lowest approximation error. After calculating the suboptimal lagrangian multiplier, we update $\Lambda$, i.e., $\Lambda_t \leftarrow \Lambda_{t-1} \cup \hat{\lambda}_t$. As illustrated in Figure 2b, when the multimodal model approaches its global minimum, the parameters of RegBN also converge.

## 4  Experiments

Heterogeneous data captured by diverse multisource sensors are utilized to verify the usefulness and effectiveness of RegBN in multiple data contexts such as language, audio, 2D image, video, depth, 3D MRI, and tabular data. RegBN is applied to eight datasets that are summarized in Table 1. Details on the datasets and baseline methods are provided in Appendices **??** & **??**. The default parameters and settings for RegBN are reported in Appendix **??**. Our code is openly available at https://mogvision.github.io/RegBN. Experimental details are provided in Appendix **??**. Here, we summarize the main results. The findings of this investigation provide insights into the performance of RegBN across various data modalities and highlight its potential as a robust normalization technique. PMDN relies on a considerable number of learnable parameters specifically tailored for metadata with limited dimensions. MDN necessitates substantial RAM resources, particularly when estimating its inverse matrix on large-scale datasets.

### 4.1  Multimedia

**Experiments with the LLP dataset [53]**: Audio-Visual Video Parsing (AVVP) [53] (see Appendix **??**) is employed as a baseline for parsing individual audio, visual, and audio-visual events under both segment-level and event-level metrics over the LLP dataset. The normalization module is employed to decouple the audio features from the video features, ensuring their independence. Table 2 reports that RegBN improves the baseline performance in nine out of ten metrics. RegBN brings about improvements in all audio-visual video parsing subtasks, measured both at the segment-level and event-level metrics. These findings suggest that decoupling audio and video allows AVVP to produce more accurate predictions of event categories on a per-snippet basis.[1]

**Experiments with the MM-IMDb dataset [3]**: The MM-IMDb dataset was curated for the purpose of predicting movie genres through the use of either image or text modality. This task involves

---

[1]Appendix **??** reports the validation results of AVVP with and without RegBN on the LLP dataset.

Table 2: Audio-visual video parsing accuracy (%) of AVVP [53], as baseline (BL), on the LLP dataset [53] for different normalization techniques. A and V stand for audio and visual, respectively.

| Method | Segment-Level | | | | | Event-Level | | | | |
|---|---|---|---|---|---|---|---|---|---|---|
| | A↑ | V↑ | A-V↑ | Type↑ | Event↑ | A↑ | V↑ | A-V↑ | Type↑ | Event↑ |
| BL | 60.1 | 52.9 | 48.9 | 54.0 | 55.4 | 51.3 | 48.9 | 43.0 | 47.7 | 48.0 |
| BL+PMDN | 59.7 | 53.0 | 48.8 | 53.6 | 55.3 | 51.4 | 49.1 | 42.9 | 47.7 | 48.2 |
| BL+RegBN | 60.2 | 53.3 | 48.9 | 54.0 | 55.2 | 52.0 | 49.5 | 43.1 | 47.9 | 49.3 |

Table 3: Multi-label classification scores (F1 score) of baseline SMIL [31] (denoted by BL) with/without normalization on the MM-IMDb dataset [3]

| Method | Norm. Params. | F1 Score (%) | | |
|---|---|---|---|---|
| | (#) | Samples↑ | Micro↑ | Weighted↑ |
| BL | – | 49.62 | 51.18 | 48.95 |
| BL+PMDN | 65,536 | 49.85 | 51.44 | 49.36 |
| BL+RegBN | 0 | 54.82 | 55.37 | 52.83 |

multi-label classification since a single movie may be associated with multiple genres (see Appendix **??**). SMIL [31], as a baseline approach, employs the pre-trained BERT to extract the textual features, while image features are extracted using the pre-trained VGG-19. The text features are subsequently normalized with respect to the visual features using the proposed RegBN before the fusion process occurs. The fusion operation involves a concatenation layer. The fused features are passed through two fully-connected layers to obtain the classification labels. The details are provided in Appendix **??**. Table 3 presents the quantitative results obtained from the experiments. The table indicates that applying a normalization layer such as PMDN or RegBN can lead to an improvement in the performance over the baseline model. This highlights the importance of the normalization step in multimedia data. In particular, RegBN was found to enhance the F1 score of the baseline by an average of 8.86% across all three metrics.

**Experiments with the AV-MNIST dataset [31]**: The small AV-MNIST dataset comprises two modalities, namely, audio and image. The audio modality is obtained from Free Spoken DigitsDataset[2], which includes 1,500 raw audio recordings from three different speakers. Despite the independence of the image and audio modalities, the primary objective of this experiment is to investigate whether the confounding effects induced by the speakers have an impact on the classification results. In line with the previous experiment, SMIL [31] serves as the baseline method. The obtained classification results for the combinations of baseline+MDN, baseline+PMDN, and baseline+RegBN are 98.43%, 98.68%, and 99.11%, respectively. Figure 3 illustrates t-SNE plots for the different approaches. In this experiment, tSNE is applied to the features extracted from a fully-connected layer after the concatenation of normalized image features with audio features. The figure demonstrates that the classes are more separable for RegBN as compared to MDN or PMDN, in particular, class '2'. This result implies that our normalization method effectively removes the confounding effects caused by different speakers, thereby providing more flexibility to the baseline method for accurate classification. The results on AV-MNIST are reported in Appendix **??**.

## 4.2 Affective computing

In this section, we explore the potential application of RegBN in multimodal emotion-sentiment analysis using the IEMOCAP [8], CMU-MOSI, and CMU-MOSEI [63] datasets, which are divided into aligned and non-aligned multimodal time-series (see Appendices **??**, **??**, & **??**). The baseline method (BL) for this task is the Multimodal Transformer (MulT) [54], which is a ViT developed for analyzing multimodal language sequences. MulT fuses video ($v$), audio ($a$), and language ($l$) at three levels (see Appendix **??**): 1) language fusion ($v \rightarrow l$ & $a \rightarrow l$), 2) audio fusion ($v \rightarrow a$ & $l \rightarrow a$), and 3) video fusion ($a \rightarrow v$ & $l \rightarrow v$). The outputs of one or more levels are then aggregated for

---

[2]Free Spoken Digit Dataset is available at https://github.com/Jakobovski/free-spoken-digit-dataset

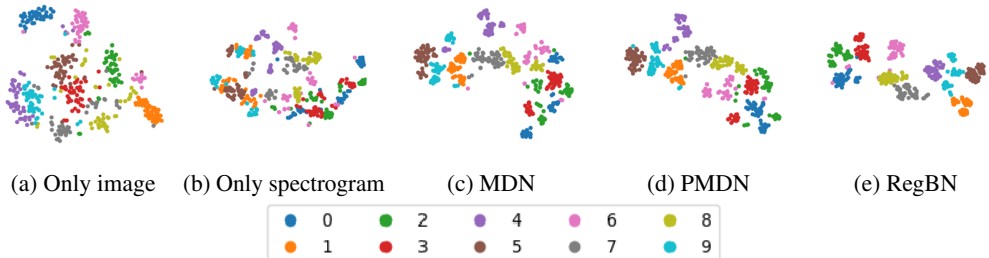

(a) Only image    (b) Only spectrogram    (c) MDN    (d) PMDN    (e) RegBN

Figure 3: tSNE visualization of the features extracted from a-b) an unimodal image and an unimodal audio, and c-e) the multimodal model with different normalization methods. Each data point represents a sample.

Table 4: Results of multimodal emotion analysis on IEMOCAP [8] with word aligned multimodal sequences. The baseline (BL) is Multimodal Transformer (MulT) [54].

| Method | Loss | | Happy (%) | | Sad (%) | | Angry (%) | | Natural (%) | |
|---|---|---|---|---|---|---|---|---|---|---|
| | Train↓ | Test↓ | Acc↑ | F1↑ | Acc↑ | F1↑ | Acc↑ | F1↑ | Acc↑ | F1↑ |
| BL | 0.106 | 0.536 | 86.3 | 84.0 | 81.5 | 80.6 | 86.5 | 86.4 | 69.5 | 69.1 |
| BL+RegBN | 0.009 | 0.452 | 87.4 | 83.0 | 84.3 | 84.1 | 88.2 | 88.1 | 73.4 | 73.2 |

classification. Here, we report the results of all three fusion levels combined, while the detailed results for each level are provided in Appendix **??**. The quantitative results of the emotion analysis experiment are reported in Table 4. The table shows that RegBN improves the classification performance of MulT in most cases, particularly for the "natural" class of the word-aligned experiment. Table 5 presents the results of the multimodal sentiment analysis on the CMU-MOSI and CMU-MOSEI datasets. In both the word-aligned and unaligned experiments, decoupling the multimodal features with RegBN improves training and inference performance.

## 4.3    Healthcare diagnosis

The ADNI dataset [19] includes 3D MRI scans of patients along with rich clinical information organized in a low-dimensional tabular format (see Appendix **??**). The dataset is subject to confounding effects such as age, sex, and level of education, which must be considered to prevent biased evaluation results, as noted in studies [59, 39]. Similar to [60], we partitioned the data into five separate and non-overlapping folds, ensuring that each fold had a balanced distribution of diagnosis, age, and sex. In this experiment, we utilized a 3D ResNet [60] for extracting MRI features, while tabular features were extracted using an MLP network proposed in [14] (see Appendix **??**). These features were then concatenated and fed into another MLP network for diagnosis classification. We applied different normalization techniques to the unimodal MRI and tabular features prior to concatenation. Table 6 reports the mean and standard deviation obtained by averaging the performance of the multimodal over these five folds. The complete results can be found in Appendix **??**. RegBN emerged as the most effective technique, outperforming other normalization methods in terms of both ACC and BA metrics. Moreover, RegBN's ability to relax the interdependencies between the different modalities facilitated lower training loss in the multimodal network.

## 4.4    Robotics

The Vision&Touch dataset [23] encompasses recordings of simulated and real robotic arms that are equipped with visual (RGB and depth), force, and proprioception sensors (see Appendix **??**). We employed 'Making Sense of Vision and Touch' (MSVT) developed by Lee et al. [23] as a baseline (refer to Appendix **??**). For this experiment, we apply normalization techniques to decouple the visual RGB and force from depth information. Table 7 summarizes the results, while detailed results can be found in Appendix **??**. The flow loss values in these tables indicate that MSVT converges more effectively when used in conjunction with RegBN. This suggests that RegBN is able to effectively remove dependencies between the RGB and depth pair, as well as the force and depth pair.

Table 5: Multimodal sentiment analysis results multimodal on CMU-MOSEI & MOSI [63] with word aligned multimodal sequences. The baseline (BL) method is MulT [54].

| Method | Dataset | Loss | | Sentiment (%) | | | | |
|--------|---------|------|------|------|------|------|------|------|
| | | Training↓ | Test↓ | Acc$_2$↑ | Acc$_5$↑ | Acc$_7$↑ | F1↑ | Corr↑ |
| BL | CMU-MOSEI | 0.452 | 0.636 | 80.3 | 51.9 | 50.3 | 80.1 | 67.1 |
| BL+RegBN | CMU-MOSEI | 0.438 | 0.611 | 81.1 | 52.2 | 50.5 | 81.2 | 66.6 |
| BL | CMU-MOSI | 0.403 | 0.632 | 81.4 | 42.5 | 37.5 | 82.0 | 69.7 |
| BL+RegBN | CMU-MOSI | 0.267 | 0.546 | 81.8 | 42.3 | 38.6 | 82.3 | 69.1 |

Table 6: Training's cross-entropy (CE) loss along test's accuracy (ACC) and balanced accuracy (BA) results on the ADNI dataset [19]. Baseline is a combination of techniques developed in [14, 60].

| Method | Norm. Params. | Training CE | Test | |
|--------|---------------|-------------|------|------|
| | (#) | loss↓ | ACC (%) ↑ | BA (%)↑ |
| BL+BN | 512 | 0.641±0.01 | 48.8±4.4 | 48.3±4.5 |
| BL+MDN | 0 | 0.619±0.03 | 50.4±4.8 | 50.1±5.6 |
| BL+PMDN | 4,096 | 0.632±0.03 | 49.7±6.2 | 49.6±7.5 |
| BL+RegBN | 0 | 0.596±0.01 | 53.0±3.1 | 52.3±3.7 |

Table 7: Results of MSVT [23] incorporating various normalization methods on Vision&Touch [23]

| Method | Norm. Params. | Training | | Test Accuracy |
|--------|---------------|----------|------|---------------|
| | (#) | Flow loss↓ | Total loss↓ | (%)↑ |
| BL | – | 0.212 | 0.563 | 86.2 |
| BL+PMDN | 131,072 | 0.194 | 0.454 | 87.9 |
| BL+RegBN | 0 | 0.052 | 0.231 | 91.5 |

Table 8: Classification accuracy results on the synthetic dataset in the presence of a confounder. Due to the randomness of the experiment, we reported the mean and std of results over 100 runs.

| Normalization | Norm. params. (#) | | Experiment I | | Experiment II | |
|---------------|------|------|------|------|------|------|
| Method | MLP | CNN | MLP | CNN | MLP | CN |
| *reference* | – | – | 87.5 | 87.5 | 75.0 | 75.0 |
| BN | 256 | 112 | 96.4±0.4 | 94.9±0.2 | 83.1±0.8 | 85.2±0.3 |
| MDN | 0 | 0 | 91.9±0.7 | 89.7±0.7 | 79.1±0.6 | 82.2±0.7 |
| PMDN | 2048 | 201,728 | 93.4±0.6 | 92.7±0.5 | 80.8±0.7 | 84.3±0.8 |
| RegBN | 0 | 0 | 87.3±0.9 | 88.2±0.8 | 76.2±0.8 | 76.8±0.9 |

### 4.5 Synthetic dataset

Drawing from the synthetic dataset in study [30], we designed a synthetic experiment for a binary classification task. Images with a size of 64×64 are split into four regions with identical Gaussians in their background. The regions along the main diagonal are linked to the class label, while the off-diagonal ones act as confounders that do not impact the true label. We created metadata of size 16 for this dataset that includes the true binary label, the confounder value, one fake label, 12 randomly generated fake features, and 1 column with a value of one. The binary classes have overlapping, so the accuracy drops based on the amount of overlapping (see Appendix **??**). In Experiment I, the theoretical maximum accuracy of classification is 87.5%, while for Experiment II, it is 75%. The base networks employed here are a simple CNN and an MLP-based neural network. The dataset and base networks are detailed in Appendices **??** and **??**, respectively. The classification results are presented in Table 8, with more detailed results available in Appendix **??**. The results reveal that conventional normalization techniques, such as BN, GN, and IN, are unable to remove the confounding effects. The performance of the base networks that employed MDN was superior compared to those with conventional normalization methods. However, MDN still exhibits shortcomings in dealing with confounding effects. Despite the incorporation of a high number of learnable parameters, PMDN was unsuccessful in eliminating confounding effects, as the inclusion of fake values in the metadata led to the deception of the multimodal model. RegBN demonstrated its ability to remove confounder effects by yielding performance closest to the theoretical maximum in both experiments.

### 4.6 Computational cost and ablation study

We employed an NVIDIA GTX 1080 Ti with 12GB VRAM for image experiments and an NVIDIA A100 with 80GB VRAM for video experiments. Since RegBN learns the projection matrix through Eqs. 4-7, it eliminates the need for learnable parameters, resulting in enhanced computational efficiency for multimodal models. During the training process, RegBN achieves a frame rate of approximately 124 fps when normalizing a pair of layers with dimensions $256 \times 1024$. In comparison, conventional normalization techniques achieve a frame rate of around 13,000 fps for the normalization of one layer of dimensions $256 \times 1024$. Regarding inference time, RegBN exclusively relies on the updated projection matrix and therefore does not require additional computation. RegBN is frequently used before the fusion blocks in multimodal models, so the training time of a few RegbBN layers is negligible. Results for the ablation study are provided in Appendix **??**.

## 5 Conclusion and future work

This study presented a novel normalization method designed for dependency and confounding removal in multimodal data and models. The experimental results demonstrated the effectiveness of RegBN in significantly enhancing the accuracy and convergence of multimodal models, rendering it a promising normalization method for multimodal heterogeneous data. The current version of RegBN employs L-BFGS on a single GPU, which will be improved in the future. Our aspiration is that this work effectively leverages new opportunities in exploring and harnessing multi-modality data within multimodal analysis.

## Acknowledgements

This work was supported by the Munich Center for Machine Learning (MCML) and the German Research Foundation (DFG). The authors gratefully acknowledge the computational and data resources provided by the Leibniz Supercomputing Centre. The authors thank Emre Kavak for his valuable input on the synthetic dataset design.

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
