# RegBN: Batch Normalization of Multimodal Data with Regularization (Supplementary Material)

## Contents

## A  Multimodal normalization and fusion

Multimodal data fusion can be accomplished at various feature levels, including low-level, high-level, or latent space. Furthermore, these features can be integrated through different fusion strategies, such as early, middle, or late fusion [59, 36]. In multimodal research, determining the optimal fusion model and feature level for fusion remains challenging. This study showed that normalizing the features extracted from heterogeneous data sources can yield better fusion results. We demonstrate the applicability of RegBN as a multimodal normalization technique in various fusion structures within multimodal models. As depicted in Figure 4, we highlight several scenarios where RegBN can be employed effectively.

- *Layer normalization*: RegBN, as a normalization method, can be applied to any pair of multimodal layers regardless of their dimension and feature types, alleviating superimposed layers (caused by confounding factors and dependencies at both low- and high-level features) and thereby enhancing their efficiency (see Figure 4a). The experiments conducted on the synthetic dataset (Section 4.5) and the healthcare diagnosis (Section 4.3) serve as illustrative instances of layer normalization.

- *Late fusion*: This category of fusion is more popular among multimodal techniques. As shown in Figure 4b, with the assistance of RegBN, the pair of output feature layers can be rendered independent, enabling the multimodal model to seamlessly combine the superimposed layers with enhanced efficiency. The utilization of RegBN in the late fusion structure can be observed in the domains of multimedia (Section 4.1), affective computing (Section 4.2), and robotics (Section 4.4).

- *Layer fusion*: RegBN also facilitates layer-to-layer fusion. In this scenario, the corresponding layers are made mutually independent through RegBN before the fusion process takes place (see Figure 4c). An example is provided with the synthetic experiment in Section 4.5.

- *Early fusion*: RegBN ensures the independence of input data before fusion in multimodal scenarios, leveraging the potential information of each source. As illustrated in Figure 4d, the early fusion strategy is recommended when the input data exhibits high correlation, as demonstrated in the healthcare diagnosis experiment outlined in Section 4.3.

## B  Solution details

To detail the solution, we first recall Eq. 3. The goal is to find the solution of the following objective:

$$\mathcal{F}(W^{(l,k)}, \lambda_+) = \left\| f^{(l)} - W^{(l,k)} g^{(k)} \right\|_2^2 + \lambda_+ \left( \left\| W^{(l,k)} \right\|_F - 1 \right). \tag{9}$$

This equation yields a closed-form solution for $W^{(l,k)}$:

$$\hat{W}^{(l,k)} = \left( g^{(k)\top} g^{(k)} + \hat{\lambda}_+ \mathbf{I} \right)^{-1} g^{(k)\top} f^{(l)}, \tag{10}$$

where $\hat{\lambda}_+$ is obtained through the following:

$$\hat{\lambda}_+ = \underset{\lambda_+}{\arg\min} \left( \left\| \left( g^{(k)\top} g^{(k)} + \lambda_+ \mathbf{I} \right)^{-1} g^{(k)\top} f^{(l)} \right\|_F - 1 \right). \tag{11}$$

One can use SVD for simplifying Eqs. 10 & 11. Let SVD decompose layer $g^{(k)}$: $\mathrm{SVD}\left(g^{(k)}\right) = U\Sigma V^* = \sum_{i=1}^{m} \sigma_i u_i v_i^*$, then the projection weights are re-written as follows

$$\hat{W}^{(l,k)} = \sum_{i=1}^{m} \frac{\sigma_i}{\sigma_i^2 + \hat{\lambda}_+} u_i v_i f^{(l)}, \tag{12}$$

where

$$\hat{\lambda}_+ = \underset{\lambda_+}{\arg\min} \left( \left\| \sum_{i=1}^{m} \frac{\sigma_i}{\sigma_i^2 + \lambda_+} u_i v_i f^{(l)} \right\|_F - 1 \right). \tag{13}$$

The L-BFGS algorithm facilitates the estimation of $\hat{\lambda}_+$ that is then inserted into Eq 12 to compute the projection matrix. In our implementation, we employed the L-BFGS solver provided by PyTorch[3]. The default settings of the L-BFGS parameters are presented in Appendix E.1.

---

[3]The L-BFGS solver can be found at `https://pytorch.org/docs/stable/generated/torch.optim.LBFGS.html`.

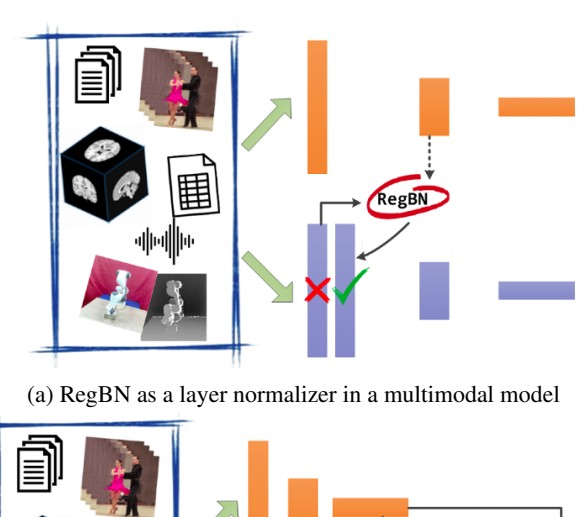

(a) RegBN as a layer normalizer in a multimodal model

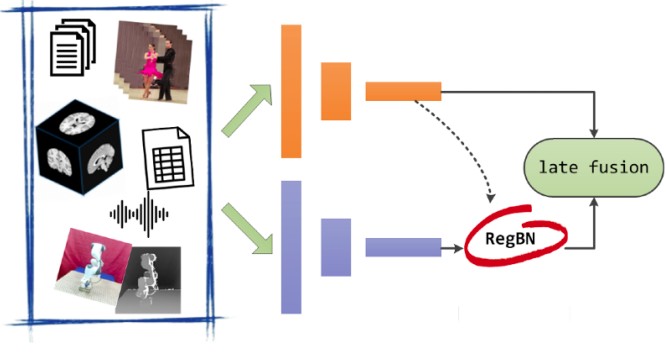

(b) Late fusion with RegBN

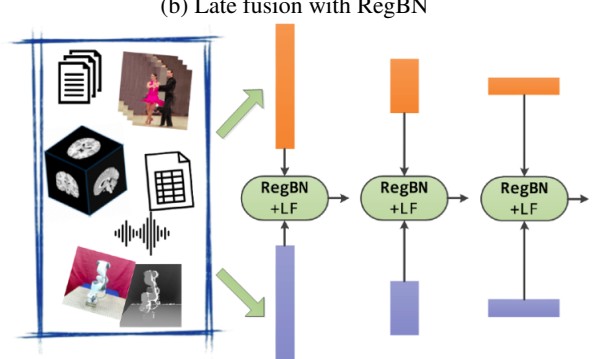

(c) Layer fusion (LF) with RegBN

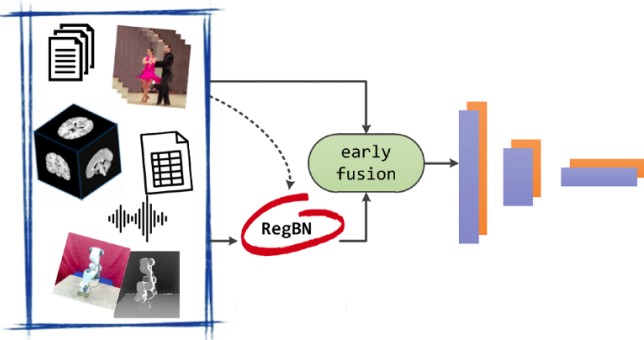

(d) Early fusion with RegBN

Figure 4: RegBN, as a multimodal normalization technique, in the context of different fusion paradigms. Given two modalities, $\mathcal{A}$ and $\mathcal{B}$, each modality has its respective pyramidal feature space (shown in different colors). (a) RegBN is capable of normalizing any pair of (input/hidden/output) layers in a multimodal neural network; (b-d) a multimodal neural network fuses the features of modalities $\mathcal{A}$ and $\mathcal{B}$ in various ways, including early, layer, and late fusion. In each fusion scenario, the inputs to the fusion block are normalized using the RegBN technique.

# C Datasets

Below, we provide a brief overview of the databases used in this study.

## C.1 LLP

"Look, Listen, and Parse" (LLP) [54] consists of 11,849 YouTube video clips, encompassing 25 event categories and totaling 32.9 hours of content sourced from AudioSet [13]. The LLP dataset includes 11,849 video-level event annotations, indicating the presence or absence of different video events to facilitate weakly-supervised learning. Each video has a duration of 10 seconds and contains at least 1 second of audio or visual events. Among the videos, 7,202 contain events from more than one event category, with an average of 1.64 different event categories per video. The samples were annotated for training with sparse labels, while the test set provides dense sound event labels for both video and audio at the frame level. According to [54], individual audio and visual events were annotated with second-wise temporal boundaries for a randomly selected subset of 1,849 videos from the LLP dataset to evaluate audio-visual scene parsing performance. The audio-visual event labels were derived from the audio and visual event labels. There are 6,626 event annotations, including 4,131 audio events and 2,495 visual events, for the 1,849 videos. Merging the individual audio and visual labels results in 2,488 audio-visual event annotations. This subset is divided into 649 videos and 1,200 videos for validation and inference, respectively. The AVVP [54] framework was trained on the 10,000 videos with weak labels, and the trained models were tested on the validation and testing sets with fully annotated labels. The audio and visual features of videos in the LLP dataset are openly accessible[4].

## C.2 MM-IMDb

The Multimodal IMDb (MM-IMDb) [3] dataset comprises two modalities, namely text and image. The aim is to perform multi-label classification on this dataset and predict the movie genre using either an image or text modality. This task is challenging as a movie can be assigned multiple genres. The dataset consists of 25,956 movies classified into 23 different genres. We use the same training and validation splits as in the previous study by [32]. The RGB images underwent standardization by rescaling from $261 \times 385$ to $256 \times 256$. They were then cropped to achieve a size of $224 \times 224$. The dataset utilized in this study is openly accessible and available for public use[5].

## C.3 AV-MNIST

AV-MNIST[6] is generated by combining spoken digit audio from the Free Spoken Digits Dataset[7] with written digits from the MNIST dataset[8]. The objective is to classify the digit into one of ten categories (0 - 9). Classification of this dataset is a challenging task as the visual modality's energy is reduced by 75% through PCA and real-world background noises are added into the audio modality. The grey images in AV-MNIST have a resolution of $28 \times 28$ pixels, while the audio spectrogram is $112 \times 112$ pixels.

## C.4 IEMOCAP

The "Interactive Emotional Dyadic Motion Capture" (IEMOCAP) dataset [8] comprises 151 videos focused on dyadic interactions for human emotion analysis. The dataset comprises around 12 hours of audiovisual data, encompassing various modalities such as video, speech, motion capture of the face, and text transcriptions. The IEMOCAP database is annotated by multiple annotators, assigning categorical labels to the data, including emotions such as anger, happiness, sadness, and neutrality, and dimensional labels such as valence, activation, and dominance. In line with study [55], four emotions were chosen, including happy, sad, angry, and neutral, which are used for emotion recognition. It is important to note that IEMOCAP is a multilabel task, meaning that multiple emotions can be assigned

---

[4] https://github.com/YapengTian/AVVP-ECCV20
[5] https://github.com/pliang279/MultiBench
[6] The dataset is available at https://github.com/pliang279/MultiBench.
[7] https://github.com/Jakobovski/free-spoken-digit-dataset
[8] http://www.pymvpa.org/datadb/mnist.html

to an individual. The dataset's multimodal streams have fixed sampling rates for audio signals at 12.5 Hz and for vision signals at 15 Hz. In line with study [55], the evaluation of the dataset involves reporting the binary classification accuracy and the F1 score of the predictions.

## C.5 CMU-MOSI

The Multimodal Opinionlevel Sentiment Intensity (CMU-MOSI) [64] is a comprehensive compilation of 2,199 short monologue video clips, each meticulously labeled with various annotations. These annotations include subjectivity labels, sentiment intensity labels, per-frame visual features, per-opinion visual features, and per-millisecond audio features. The CMU-MOSI dataset serves as a realistic and practical multimodal dataset for the task of affect recognition. It is widely utilized in competitions and workshops focusing on affect recognition research. The preprocessed versions of the CMU-MOSI dataset are openly accessible[9]. The annotation of sentiment intensity within the dataset encompasses a comprehensive range from -3 to +3. This extensive annotation range enables the development of fine-grained sentiment prediction capabilities that go beyond the conventional positive/negative categorization, facilitating a more nuanced understanding of sentiment in the data. The videos within the dataset feature a diverse group of 89 speakers, with a distribution of 41 female speakers and 48 male speakers. These speakers represent a range of backgrounds, including Caucasian, African-American, Hispanic, and Asian individuals. Most speakers fall within the age range of 20 to 30 years. It is worth noting that all speakers in the dataset express themselves in English, and the videos originate from either the United States of America or the United Kingdom. Age, gender, and race are confounding factors. The training set consists of 52 videos, the validation set contains 10 videos, and the test set comprises 31 videos. Splitting these videos into segments results in a total of 1,284 segments in the training set, 229 segments in the validation set, and 686 segments in the test set.

## C.6 CMU-MOSEI

CMU Multimodal Opinion Sentiment and Emotion Intensity (CMU-MOSEI)[10] is a large-scale dataset specifically designed for sentence-level sentiment analysis and emotion recognition in online videos, comprising over 65 hours of annotated video data sourced from a diverse set of over 1,000 speakers and covering more than 250 topics. Videos in the CMU-MOSEI dataset were meticulously annotated for sentiment, along with identifying nine distinct emotions: anger, excitement, fear, sadness, surprise, frustration, happiness, disappointment, and neutrality. Continuous emotions such as valence, arousal, and dominance were also annotated. The inclusion of diverse prediction tasks makes CMU-MOSEI a highly valuable dataset for evaluating multimodal models across various affective computing tasks encountered in real-world scenarios. This experiment divided the dataset into three subsets: training, validation, and test sets. The training set contained 16,265 samples, the validation set had 1,869 samples, and the test set comprised 4,643 samples. The dimensions of the text data, audio data, and vision data are as follows: the text data has dimensions of $50 \times 300$, the audio data has dimensions of $500 \times 74$, and the vision data has dimensions of $500 \times 35$. The accuracy of the results with the CMU-MOSEI and CMU-MOSI datasets is assessed using the following evaluation metrics:

- Binary accuracy ($Acc_2$): This metric indicates the accuracy of sentiment classification as either positive or negative.
- 5-Class accuracy ($Acc_5$): In this evaluation, each sample is labeled by human annotators with a sentiment score ranging from -2 (indicating strongly negative) to 2 (representing strongly positive).
- 7-Class accuracy ($Acc_7$): Similar to $Acc_5$, each sample is labeled with a sentiment score ranging from -3 (strongly negative) to 3 (strongly positive).

## C.7 ADNI

The "Alzheimer's Disease Neuroimaging Initiative" (ADNI)[11] database is the most popular benchmark for Alzheimer's research and diagnosis. The dataset includes both 3D MRI scans and tabular metadata.

---

[9]https://github.com/pliang279/MultiBench
[10]https://github.com/A2Zadeh/CMU-MultimodalDataSDK
[11]https://adni.loni.usc.edu

The objective is to categorize patients into three groups: cognitively normal (CN), mildly cognitively impaired (MCI), or Alzheimer's demented (AD). The dataset was prepared in line with [61]. T1-weighted MRIs were first normalized with minimal pre-processing and then segmented using the FreeSurfer v5.3 software[12]. Only the regions of size $64 \times 64 \times 64$ around the left hippocampus were extracted since this region is strongly affected by Alzheimer's disease. The tabular data as metadata comprises nine variables that contribute valuable information to the dataset. These variables include ApoE4, which indicates the presence or absence of the Apolipoprotein E4 allele known to be associated with an elevated risk of Alzheimer's disease. The dataset also contains variables related to cerebrospinal fluid biomarkers, P-tau181 and T-tau, which provide insights into the pathological changes associated with the disease. Additionally, demographic variables such as age, gender, and education are included. Two derived measures, obtained from 18F-fluorodeoxyglucose (FDG) and florbetapir (AV45) PET scans, serve as summary measures in the dataset. In accordance with the methodology presented in the work by Wolf et al. [61], the dataset underwent a partitioning process into five distinct folds. This partitioning strategy was carefully devised to ensure a balanced distribution of diagnosis, age, and sex across the folds. During the evaluation process, each of the five folds was used as a test set once, ensuring comprehensive coverage across the dataset. The remaining folds were further divided into five equally balanced chunks. From these chunks, one chunk was randomly selected to serve as the validation set, while the remaining data within the folds constituted the training set. As a result, the resulting distribution of data splits consists of 20% for the test set, 16% for the validation set, and 64% for the training set. This partitioning scheme aims to maintain a proportional and representative distribution of the data subsets, facilitating reliable evaluation and training procedures.

## C.8 Vision&Touch

Vision&Touch, introduced by Lee et al. [24], is a collection of real-world robot manipulation data. It encompasses visual, force, and robot proprioception information. The data is obtained by executing two policies on the robot: a random policy that takes random actions and a heuristic policy that aims to perform peg insertion. The dataset includes several sensor modalities: robot proprioception, RGB-D camera images, and force-torque sensor readings. The proprioceptive input comprises the pose of the robot's end-effector, as well as linear and angular velocity. RGB images and depth maps are captured using a fixed camera, which is positioned to focus on the robot. The force sensor provides feedback on six axes, measuring the forces and moments along the x, y, and z axes. The primary objective of the dataset was to support representation learning specifically for reinforcement learning applications. The Vision&Touch dataset was split into training, validation, and testing with 36,499, 4,047, and 22,800 samples, respectively. The dimensions of the RGB and depth images are $128 \times 128 \times 3$ and $128 \times 128 \times 1$, respectively. Data of size $6 \times 32$ and $8$ is acquired from force-torque sensors and robot proprioception, respectively. The database is publicly accessible[13].

## C.9 Synthetic dataset

Inspired by [31], we designed this dataset to evaluate a confounding variable's influence on the training and inference procedures of multimodal models. The primary objective of the synthetic dataset is a binary classification of two distinct groups of data, called Group 1 and Group 2. Each group consists of a collection of 5,000 images, each with a resolution of $64 \times 64$ pixels. Each image was partitioned into four quadratic sub-images (i.e., $32 \times 32$) by dividing the input image horizontally and vertically into two equal halves. All the sub-images were generated using a 2D Gaussian distribution with a standard deviation of 5. The magnitudes of sub-images in the main diagonal were multiplied by $\sigma_{cls}$, corresponding to the classification label. Similarly, the magnitudes of the bottom-left quadrant sub-image were multiplied by $\sigma_c$, which plays the role of a confounding factor. In line with the methodology proposed in [31], the magnitudes of the top-right sub-image were multiplied by zero to simplify the experiment. The values of $\sigma_{cls}$ for images in Group 1 were randomly sampled from a uniform distribution $\mathcal{U}(1, 5)$. Likewise, for images in Group 2, $\sigma_{cls}$ values were randomly sampled from a uniform distribution $\mathcal{U}(4, 8)$. Due to the overlapping of labels between the two groups, the maximum achievable accuracy, in theory, is 87.5%. Regarding the confounding variable $\sigma_c$, it was assigned a random number within the same range as the true label, i.e.,

---

[12]https://surfer.nmr.mgh.harvard.edu/fswiki/DownloadAndInstall5.3
[13]https://sites.google.com/view/visionandtouch

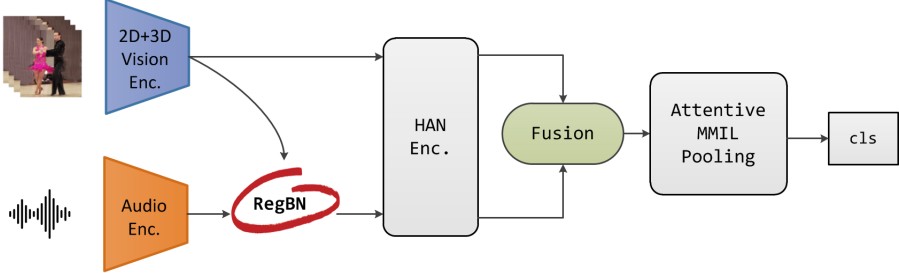

Figure 5: The inference framework of AVVP [54] with RegBN as a multimodal normalization method

$\sigma_c \in \mathcal{U}(1,5)$ for Group 1 and $\sigma_c \in \mathcal{U}(4,8)$ for Group 2. In addition to the images, metadata of length 16 was created. This metadata comprises the (true) binary label, the confounding variable $\sigma_c$ value, one randomly generated fake binary label, twelve randomly generated floating-point fake features, and one column with a constant value of one. In the ideal scenario, a multimodal model should primarily consider the first column of the metadata for classification since it uses a loss cross-entropy function with regard to the true labels during training, and the values in the other columns do not provide significant information, just used as metadata. Furthermore, the experiment was repeated for other values, referred to as Experiment II, where $\sigma_{cls}$ and $\sigma_c$ were uniformly sampled from the range $\mathcal{U}(1,7)$ for Group 1 and $\mathcal{U}(4,10)$ for Group 2. In Experiment II, the theoretical maximum accuracy is 75.0%.

## D  Baseline methods with RegBN as multimodal normalization method

Here, we review the baseline methods that are employed in this study while also detailing the application of the proposed normalization method in the normalization of the multimodal data.

### D.1  AVVP

Tian et al. [54] proposed Audio-Visual Video Parsing (AVVP) for parsing individual audio, visual, and audio-visual events. The framework of AVVP is illustrated in Figure 5. Visual and snippet-level features are extracted from 2D frame-level and 3D snippet-level features using ResNet152[14] and ResNet (2+1)D[15], respectively. Likewise, audio features are extracted via VGGish[16]. Audio, visual, and snippet-level feature dimensions are $10 \times 128$, $80 \times 2048$, and $10 \times 512$, respectively. The normalization module is employed to decouple the audio features from the visual ones, ensuring their independence. The AVVP method employs a hybrid attention network (HAN) to predict both audio and visual event labels based on the aggregated features. The HAN block incorporates both a self-attention network and a cross-attention network, enabling the HAN to dynamically learn which bimodal and cross-modal snippets to look for in every audio or visual snippet. An attentive multimodal multiple-instance learning (MMIL) pooling technique is employed to facilitate adaptive prediction of video-level event labels in weakly-supervised learning. This approach allows for adaptively pooling information from multiple instances in a multimodal manner. Additionally, an individual-guided learning strategy addresses the modality bias problem, ensuring fair representation and consideration of each modality in the learning process. For the final classification step, a fully-connected layer is utilized, followed by a sigmoid activation layer.

### D.2  SMIL

SMIL [32][17] stands for multimodal learning with severely missing modality, a method for learning a multimodal model from a complete or an incomplete dataset. The framework of SMIL is illustrated

---

[14]https://pytorch.org/vision/main/models/generated/torchvision.models.resnet152.html
[15]https://pytorch.org/vision/main/models/generated/torchvision.models.video.r2plus1d_18.html
[16]https://github.com/tensorflow/models/tree/master/research/audioset/vggish
[17]https://github.com/mengmenm/SMIL

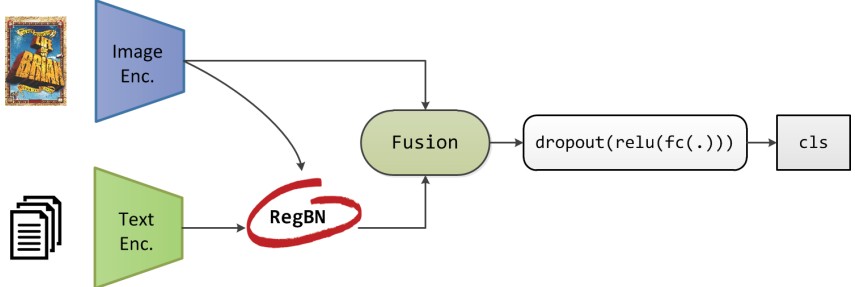

Figure 6: The inference framework of SMIL [32] with RegBN as a multimodal normalization method on MM-IMDb

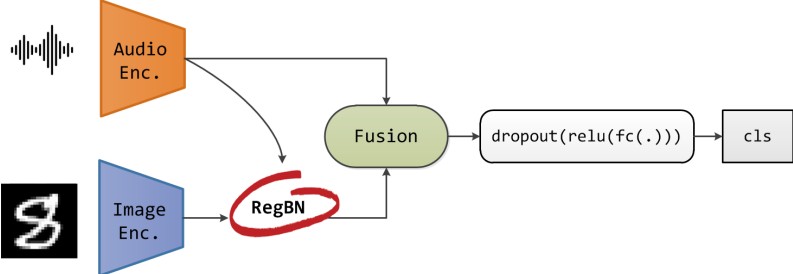

Figure 7: The inference framework of SMIL [32] with RegBN as a multimodal normalization method on AV-MNIST

in Figures 6&7. For the MM-IMDb dataset (Figure 6), the textual data is converted to lowercase, followed by feature extraction using the pre-trained BERT[18,19] models. The length of text features is 768. Features of length 512 were extracted from the images utilizing the pre-trained VGG-19[20] model. The text features are subsequently normalized with respect to the visual features using the proposed RegBN before the fusion process occurs. The fusion operation involves a concatenation layer. The fused features are passed through two fully-connected layers to obtain the classification labels. Likewise, a similar framework is employed for the audio-vision classification of the AV-MNIST dataset. As shown in Figure 7, modified LeNet-5 (refer to [32] for details) and LeNet-5 models are employed for extracting features, audio, and images, respectively. The feature dimension is 192 for audio features and 48 for vision features. We normalize the vision features with regard to the audio ones. Subsequently, the normalized vision features are concatenated with audio features and then fed into two fully-connected layers for classification.

## D.3 MulT

MulT, introduced by Tsai et al. [55] in their work on multimodal language sequences, is a transformer-based model designed specifically for multimodal data representation. As shown in the framework of MulT in Figure 8, this technique combines multimodal time series, including language/text ($l$), video/vision ($v$), and audio ($a$) modalities, through a feed-forward fusion mechanism using multiple directional pairwise crossmodal transformers. MulT has been developed to address the complexities associated with multimodal language sequences, which often exhibit an unaligned nature and require the inference of long-term dependencies across modalities. MulT is designed to handle both word-aligned and unaligned versions of these sequences. At a high-level feature, MulT effectively merges and integrates information from different modalities to capture the complex relationships within the multimodal data. Each crossmodal transformer in MulT serves to reinforce a target modality with low-level features from another modality by learning their attention-based interactions. This is done

---

[18]https://github.com/google-research/bert
[19]https://huggingface.co/docs/transformers/index
[20]https://pytorch.org/vision/main/models/generated/torchvision.models.vgg19.html

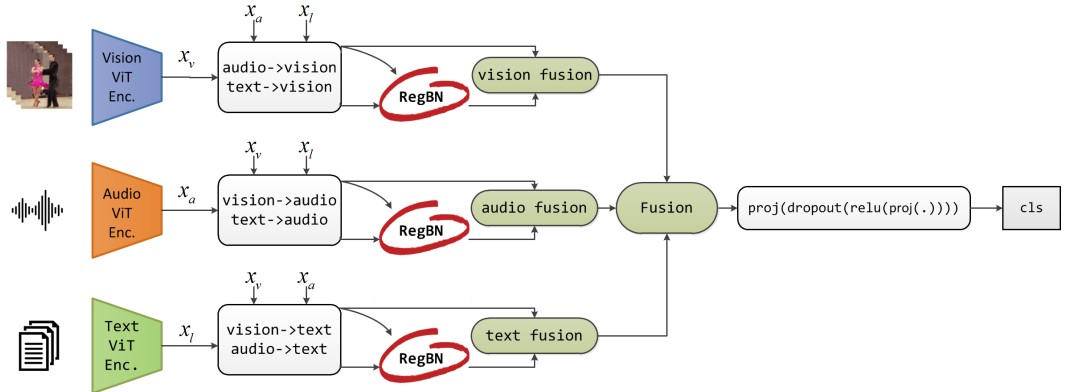

Figure 8: The inference framework of MulT [55] with RegBN as a multimodal normalization method

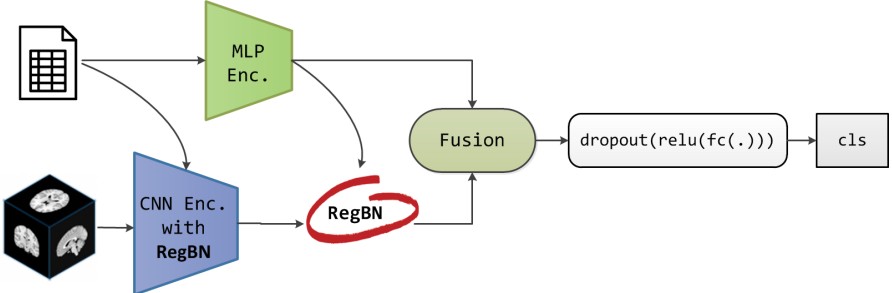

Figure 9: The framework of healthcare diagnosis model with RegBN as a multimodal normalization method

in three levels, including 1) language fusion ($v \to l$ & $a \to l$), 2) audio fusion ($v \to a$ & $l \to a$), and 3) video fusion ($a \to v$ & $l \to v$). The MulT architecture encompasses crossmodal transformers that model interactions between all pairs of modalities. This is followed by sequence models, such as self-attention transformers, which use fused features for prediction. Before each level of fusion, RegBN is performed on the pair of multimodal with high-level features as a normalization block.

## D.4 Healthcare model

For the healthcare experiment, which is the diagnosis of Azhimere on the ADNI dataset, we utilize the 3D ResNet and MLP network proposed in studies by Wolf et al. [61] and Hager et al. [15], respectively. These networks extract 3D MRI and tabular features, respectively. As shown in Figure 9, we apply normalization techniques to the abovementioned models in low- and high-level features. The 3D ResNet consists of an input convolution layer and four ResNet blocks. During feature extraction from MRI data, the features undergo normalization using a technique listed in Table 15. The output feature vector from the 3D ResNet is of length 256. On the other hand, the MLP network applies a fully-connected layer, followed by a ReLU activation layer, three times to process the tabular data. The output feature vector from the MLP network is of length 16. Once again, the visual and tabular data are normalized at the high-level feature stage. This is followed by a combination of dropout, ReLU activation, and fully-connected layers before being fed into a classification block.

## D.5 MSVT

Making Sense of Vision and Touch (MSVT) [24] is a self-supervised multimodal approach aimed at acquiring a representation of sensory multimodal inputs. This approach involves training an end-to-end representation learning network through self-supervision. As shown in Figure 10, the MSVT method utilizes data from four distinct sensors: RGB images, depth maps, force-torque readings within a 32ms timeframe, and the position, orientation, and velocity of the robot's end-

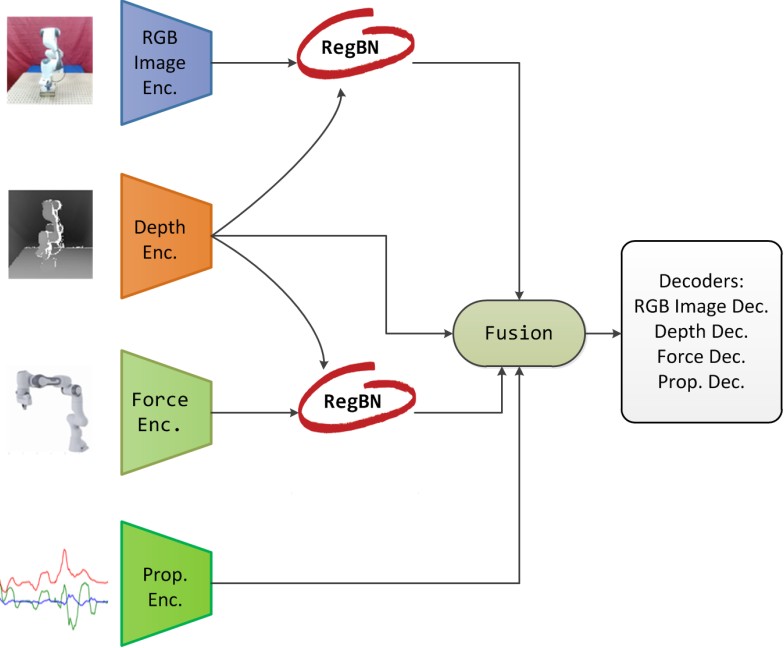

Figure 10: The framework of the MSVT model with RegBN as a multimodal normalization method

effector. These data inputs are encoded and fused into a multimodal representation using a variational Bayesian technique, enabling the learning of a policy for manipulation tasks involving contact-rich environments.

For the RGB images, a neural network with a convolutional neural network (CNN) backbone is employed. The CNN architecture comprises six convolutional layers with downsampling ($3 \rightarrow 16 \rightarrow 32 \rightarrow 64 \rightarrow 64 \rightarrow 128 \rightarrow 128$). Subsequently, the extracted features are flattened and passed through a fully-connected layer, resulting in a feature vector of length 256. Similarly, the depth data is processed using six convolutional layers with downsampling and one fully-connected layer. The output dimension of the depth features is 256. To account for the correlation between visual and depth data, the RGB image is subjected to normalization with respect to the depth image using the RegBNat technique at higher levels of feature representation. The force encoder module comprises five convolutional units, each consisting of a 1D convolution layer with downsampling followed by a LeakyReLU activation function with a negative slope of 0.1 ($6 \rightarrow 16 \rightarrow 32 \rightarrow 64 \rightarrow 128 \rightarrow 256$). The force encoder maps input data of size $32 \times 6$ to a feature vector of 256, capturing the sliding contact dynamics along the x, y, and z axes, which are superimposed on the depth information. To address this overlap, the force sensor data is normalized with respect to the depth features. The proprioception encoder module is composed of four blocks, with each block containing a fully-connected layer followed by a LeakyReLU activation function with a negative slope of 0.1 ($8 \rightarrow 32 \rightarrow 64 \rightarrow 128 \rightarrow 256$). The resultant of all the encoders are concatenated/fused. The reconstruction decoders are similar to the encoders. More details can be found in [24][21].

### D.6 CNN and MLP networks for the synthetic experiment

Due to the simple structure of the synthetic data, MLP and CNN models developed for this experiment are relatively light. The MLP model incorporates three consecutive convolution layers, without normalization, to process input $64 \times 64$ synthetic images. These convolution layers are followed by a fully-connected layer, which converts the features into a feature vector of length 128. Subsequently, the resultant vector is normalized with respect to the input tabular data using one of the techniques

---

[21]The code of MSVT is available at `https://github.com/stanford-iprl-lab/multimodal_ representation`.

Table 9: The default setting for RegBN's parameters

| parameter | value |
|---|---|
| **Parameters of exponential moving average** | |
| decay rate $\beta_1$ in Eq. 6 | 0.9 |
| decay rate $\beta_2$ in Eq. 6 | 0.99 |
| $\Lambda_p$ in Eq. 8 | [1, 100, 1000] |
| | |
| **Parameters of the L-BFGS solver** | |
| learning rate | 1.0 |
| maximal number of iterations per optimization step | 25 |
| termination tolerance on first-order optimality | 0.00001 |

mentioned in Table 17. Consequently, the normalized feature vector is passed through a fully-connected layer with a sigmoid activation function for binary classification. The CNN architecture is structured as follows for processing input synthetic images: Four convolutional layers are applied, each with a stride of 2. Each convolutional layer is equipped with a normalization technique mentioned in Table 17, followed by a ReLU activation function. The output from the convolutional layers is then flattened to convert it into a one-dimensional feature vector. This feature vector is then passed through a fully-connected layer with a ReLU activation function. Finally, another fully-connected layer with a sigmoid activation function is applied for binary classification.

# E   Experimental Results and Details

This section presents and provides a comprehensive account of the results that were not included in the paper due to page limit constraints. It is worth noting that we used the default settings of different normalization methods recommended. The batch size for most methods, including RegBN, was set to 50 in all the experiments conducted. However, it should be noted that the batch size for MDN was set to 200, which differs from the other methods.

## E.1   RegBN parameter setting

To update its learnable projection matrix, RegBN employed the exponential moving average method and the L-BFGS optimizer that incorporates specific parameters (see Sections 3.1&3.2). Table 9 presents the predetermined parameters along with their respective constant values, which remained unchanged throughout the entirety of the experiment conducted in this study. In Section F, we present an ablation study on the predetermined parameters of RegBN.

## E.2   Multimedia

The validation results of AVVP with the LLP dataset are presented in Table 10. We report the validation results to show how AVVP can be efficiently trained in the presence of RegBN. RegBN improves the segment-level and event-level results of AVVP for audio (A), vision (V), and audio-vision (A-V) inputs, enhancing the performance across all modalities. Table 11 showcases the classification accuracy achieved by SMIL on the AV-MNIST dataset. The table clearly demonstrates that incorporating multiple modalities leads to superior results compared to using individual modalities alone. By utilizing RegBN as a multimodal normalization technique, the performance of the baseline SMIL model is improved. This highlights the importance of normalizing multimodal data prior to fusion, emphasizing the necessity of such normalization for gaining better results.

## E.3   Affective computing

Detailed results of MulT with and without RegBN for the IEMOCAP, CMU-MOSEI, and CMU-MOSI datasets can be found in Tables 12, 13, and 14, respectively. These tables comprehensively analyze the impact of incorporating RegBN in the MulT model. RegBN consistently improves the performance of MulT across various fusion categories. In both solo fusion and all fusion categories, RegBN enhances the results of MulT in most cases. Notably, significant improvements can be observed when using RegBN for audio or video fusion of word-aligned CMU-MOSI and for language fusion of IEMOCAP

Table 10: Audio-visual video parsing's *validation* accuracy (%) of baseline AVVP [54], as baseline (BL), on the LLP dataset [54] for different normalization techniques. AVVP has 4,571,723 learnable parameters. PMDN requires 26,214,400 parameters in its architecture, while RegBN does not need any learnable parameters as its learnable parameters are learned in a self-supervised way, described in Section 3.

| Method | Segment-Level | | | | | Event-Level | | | | |
|---|---|---|---|---|---|---|---|---|---|---|
| | A↑ | V↑ | A-V↑ | Type↑ | Event↑ | A↑ | V↑ | A-V↑ | Type↑ | Event↑ |
| BL | 61.8 | 54.5 | 49 | 55.1 | 57.4 | 53.6 | 49.9 | 43.3 | 49.4 | 49.8 |
| BL+PMDN | 61.4 | 54.6 | 48.9 | 54.8 | 57.4 | 52.8 | 50.5 | 43.3 | 49.3 | 50.2 |
| BL+RegBN | 63.5 | 55.3 | 49.1 | 55 | 58 | 52.5 | 51.1 | 44 | 49 | 50.9 |

Table 11: Classification accuracy of baseline SMIL [32] (denoted by BL) with/without normalization on the AV-MNIST dataset

| Method | Norm. params. (#) | ACC(%) ↑ |
|---|---|---|
| **Unimodal data** | | |
| BL (Unimodal $i$) | - | 64.3 |
| BL (Unimodal $a$) | - | 43.2 |
| | | |
| **Multimodal data** | | |
| BL | - | 70.6 |
| BL+PMDN | 11,136 | 70.7 |
| BL+RegBN | 0 | 71.1 |

with word-aligned data. Furthermore, the inclusion of RegBN leads to substantial improvements in training and test loss values, demonstrating its effectiveness in enhancing the convergence of the multimodal model. The improved loss values indicate that RegBN plays a crucial role in optimizing the training process and ensuring better generalization performance during testing.

## E.4 Healthcare diagnosis

Table 15 reports the performance of the baseline multimodal models developed in [15, 61] with different normalization techniques. Healthcare data is most often accompanied by confounding effects, and Table 15 suggests that conventional normalization methods such as BN, IN, and GN may not be suitable for effectively normalizing multimodal data. Multimodal data requires dedicated and specialized normalization methods tailored to their unique characteristics. Both MDN and PMDN demonstrate superior performance compared to conventional normalization techniques. However, the results are significantly improved when incorporating RegBN in terms of test accuracy and training loss. The RegBN results emphasize that it can effectively handle the complexities of multiple modalities.

## E.5 Robotics

As mentioned in Appendix D.5 and depicted in Figure 10, a pair of normalization units are employed for decoupling the RGB image-depth and force-depth pairs. These units are referred to as *vision* and *touch*, respectively, and the results are provided in Table 16. The observed decrease in training loss values and simultaneous increase in test accuracy clearly indicate the necessity of the mentioned normalization units.

## E.6 Synthetic dataset

Table 17 reports the accuracy results on the synthetic dataset for different batch normalization methods. The results obtained in this section are consistent with those reported in healthcare experiments. Just like in healthcare data analysis, where the multimodal data's confounding effects and unique characteristics pose challenges for conventional normalization methods, the findings in

Table 12: Detailed results of multimodal emotion analysis on IEMOCAP with **word-aligned** and **non-aligned** (inside parentheses) multimodal sequences. The baseline (BL) is MulT [55].

| Method | Loss | | Happy (%) | | Sad (%) | | Angry (%) | | Natural (%) | |
|---|---|---|---|---|---|---|---|---|---|---|
| | Training↓ | Test↓ | Acc↑ | F1↑ | Acc↑ | F1↑ | Acc↑ | F1↑ | Acc↑ | F1↑ |
| *Language Fusion ($v \rightarrow l$ & $a \rightarrow l$)* | | | | | | | | | | |
| BL | 0.141 | 0.458 | 85.6 | 84.1 | 83.1 | 81.8 | 83.8 | 83.1 | 67.5 | 66.6 |
| | (0.381) | (0.63) | (86.1) | (80.6) | (79.9) | (77.2) | (76.4) | (70.2) | (59.6) | (51.8) |
| BL+RegBN | 0.152 | 0.439 | 86.3 | 81.9 | 84.8 | 83.6 | 87.4 | 87.5 | 70.8 | 70.2 |
| | (0.335) | (0.587) | (86.1) | (80.6) | (79.8) | (76.9) | (76.0) | (70.7) | (60.0) | (55.6) |
| *Audio Fusion ($v \rightarrow a$ & $l \rightarrow a$)* | | | | | | | | | | |
| BL | 0.154 | 0.450 | 87.1 | 83.3 | 83.6 | 82.9 | 83.8 | 82.8 | 69.2 | 69.2 |
| | (0.394) | (0.678) | (86.2) | (80.8) | (79.6) | (77.0) | (76.2) | (69.9) | (60.2) | (55.2) |
| BL+RegBN | 0.009 | 0.425 | 85.8 | 82.8 | 83.6 | 83.2 | 86.0 | 85.3 | 70.1 | 67.5 |
| | (0.330) | (0.623) | (86.4) | (80.6) | (79.8) | (76.9) | (76.3) | (70.1) | (59.7) | (54.5) |
| *Video Fusion ($a \rightarrow v$ & $l \rightarrow v$)* | | | | | | | | | | |
| BL+Identity | 0.141 | 0.498 | 87.2 | 86.1 | 83.9 | 83.9 | 85.0 | 84.8 | 70 | 69.7 |
| | (0.33) | (0.669) | (86.1) | (80.6) | (80.0) | (76.9) | (76.4) | (70.2) | (59.6) | (56.2) |
| BL+RegBN | 0.189 | 0.477 | 86.6 | 84.5 | 81.2 | 79.3 | 83.1 | 82.5 | 68.7 | 68.4 |
| | (0.285) | (0.645) | (86.2) | (80.5) | (79.7) | (76.8) | (76.3) | (70) | (59.8) | (54.7) |
| *All Fusion Categories* | | | | | | | | | | |
| BL | 0.106 | 0.536 | 86.3 | 84.0 | 81.5 | 80.6 | 86.5 | 86.4 | 69.5 | 69.1 |
| | (0.307) | (0.665) | (85.1) | (80.4) | (79.6) | (77.0) | (76.4) | (70.2) | (59.8) | (55.6) |
| BL+RegBN | 0.009 | 0.452 | 87.4 | 83.0 | 84.3 | 84.1 | 88.2 | 88.1 | 73.4 | 73.2 |
| | (0.292) | (0.641) | (86.1) | (80.6) | (79.7) | (76.8) | (76.3) | (70.1) | (59.8) | (54.9) |

Table 13: Multimodal sentiment analysis results multimodal on CMU-MOSEI with **word aligned** and **non-aligned** (inside parentheses) multimodal sequences. The baseline (BL) method is MulT [55].

| Fusion | Loss | | Sentiment (%) | | | | |
|---|---|---|---|---|---|---|---|
| | Training↓ | Test↓ | $Acc_2$↑ | $Acc_5$↑ | $Acc_7$↑ | F1↑ | Corr↑ |
| *Language Fusion ($v \rightarrow l$ & $a \rightarrow l$)* | | | | | | | |
| BL | 0.463 | 0.598 | 80.9 | 51.2 | 49.7 | 81.1 | 67.3 |
| | (0.435) | (0.618) | (81.0) | (51.1) | (49.5) | (81.0) | (67.2) |
| BL+RegBN | 0.435 | 0.592 | 80.9 | 51.7 | 50.3 | 81.2 | 66.4 |
| | (0.387) | (0.600) | (81.0) | (51.8) | (50.5) | (81.2) | (67.7) |
| *Audio Fusion ($v \rightarrow a$ & $l \rightarrow a$)* | | | | | | | |
| BL | 0.480 | 0.625 | 81.3 | 51.0 | 49.5 | 81.3 | 66.3 |
| | (0.488) | (0.608) | (81.7) | (50.7) | (49.2) | (81.9) | (67.7) |
| BL+RegBN | 0.462 | 0.629 | 81.6 | 51.3 | 49.8 | 81.7 | 66.7 |
| | (0.496) | (0.641) | (82.5) | (51.5) | (49.8) | (82.0) | (66.5) |
| *Video Fusion ($a \rightarrow v$ & $l \rightarrow v$)* | | | | | | | |
| BL | 0.451 | 0.632 | 80.8 | 52.1 | 50.7 | 81.0 | 65.8 |
| | (0.48) | (0.617) | (80.6) | (50.2) | (48.9) | (80.9) | (65.7) |
| BL+RegBN | 0.418 | 0.626 | 80.7 | 51.2 | 49.8 | 81.4 | 67.0 |
| | (0.465) | (0.615) | (81.4) | (51.9) | (50.4) | (81.7) | (68.0) |
| *All Fusion Categories* | | | | | | | |
| BL | 0.452 | 0.636 | 80.3 | 51.9 | 50.3 | 80.1 | 67.1 |
| | (0.481) | (0.619) | (81) | (51.4) | (49.7) | (81.2) | (67.5) |
| BL+RegBN | 0.438 | 0.611 | 81.1 | 52.2 | 50.5 | 81.24 | 66.6 |
| | (0.453) | (0.605) | (81.4) | (52.5) | (51.2) | (81.6) | (68.3) |

Table 14: Multimodal sentiment analysis results multimodal on CMU-MOSI with **word aligned** and **non-aligned** (inside parentheses) multimodal sequences. The baseline (BL) method is MulT [55].

| Fusion | Loss | | Sentiment (%) | | | | |
|---|---|---|---|---|---|---|---|
| | Training↓ | Test↓ | $Acc_2$↑ | $Acc_5$↑ | $Acc_7$↑ | F1↑ | Corr↑ |
| | | Language Fusion ($v \to l$ & $a \to l$) | | | | | |
| BL | 0.524 | 0.480 | 79.8 | 42.3 | 36.3 | 79.7 | 65.5 |
| | (0.541) | (0.550) | (78.3) | (42.2) | (35.9) | (78.6) | (59.4) |
| BL+RegBN | 0.432 | 0.461 | 79.7 | 41.9 | 35.5 | 80.2 | 66.6 |
| | (0.409) | (0.465) | (79.3) | (40.6) | (34.3) | (79.2) | (66.3) |
| | | Audio Fusion ($v \to a$ & $l \to a$) | | | | | |
| BL | 0.508 | 0.501 | 79.6 | 44.5 | 38.1 | 81.1 | 67.2 |
| | (0.421) | (0.543) | (77.8) | (40.3) | (35.2) | (77.5) | (63.1) |
| BL+RegBN | 0.376 | 0.531 | 80.7 | 51.3 | 49.8 | 81.4 | 66.6 |
| | (0.329) | (0.495) | (80.2) | (40.9) | (35.4) | (80.5) | (65.9) |
| | | Video Fusion ($a \to v$ & $l \to v$) | | | | | |
| BL | 0.447 | 0.535 | 80.4 | 44.8 | 41.2 | 80.4 | 66.7 |
| | (0.340) | (0.537) | (77.4) | (42.2) | (37.0) | (77.1) | (63.4) |
| BL+RegBN | 0.403 | 0.524 | 80.7 | 51.2 | 49.8 | 81.4 | 67.0 |
| | (0.321) | (0.525) | (79.1) | (40.6) | (36.4) | (78.9) | (67.1) |
| | | All Fusion Categories | | | | | |
| BL | 0.403 | 0.632 | 81.4 | 42.5 | 37.5 | 82.0 | 69.7 |
| | (0.431) | (0.520) | (79.5) | (42.0) | (38.9) | (80.7) | (67.3) |
| BL+RegBN | 0.267 | 0.546 | 81.8 | 42.3 | 38.6 | 82.3 | 69.1 |
| | (0.401) | (0.481) | (81.5) | (42.9) | (39.6) | (82.0) | (68.2) |

Table 15: Training cross-entropy (CE) loss, test accuracy (ACC), and test balanced accuracy (BA) on the ADNI dataset [20]. Baseline is a combination of techniques developed in [15, 61].

| Method | Norm. Params. | Training CE | Test | |
|---|---|---|---|---|
| | (#) | loss↓ | ACC (%) ↑ | BA (%)↑ |
| BL+BN | 512 | 0.641±0.01 | 48.8±4.4 | 48.3±4.5 |
| BL+GN | 512 | 0.649±0.01 | 48.4±4.6 | 47.9±4.5 |
| BL+LN | 512 | 0.646±0.01 | 48.5±4.4 | 48.1±4.5 |
| BL+IN | 512 | 0.647±0.01 | 48.2±4.5 | 48.0±4.4 |
| BL+MDN | 0 | 0.619±0.03 | 50.4±4.8 | 50.1±5.6 |
| BL+PMDN | 4,096 | 0.632±0.03 | 49.7±6.2 | 49.6±7.5 |
| BL+RegBN | 0 | 0.596±0.01 | 53.0±3.1 | 52.3±3.7 |

Table 16: Training and test results of MSVT [24], as the baseline (BL) method, with different normalization methods on Vision&Touch [24]. *vision* and *touch* refer to the normalization of RGB image-depth and force-depth pairs, respectively.

| Method | Normalisation parameters (#) | Training | | Test Accuracy (%) |
|---|---|---|---|---|
| | | flow loss | total loss | |
| BL | – | 0.212 | 0.563 | 86.22 |
| BL+PMDN (*vision*) | 65,536 | 0.194 | 0.516 | 87.52 |
| BL+PMDN (*vision + touch*) | 131,072 | 0.194 | 0.454 | 87.94 |
| BL+RegBN (*vision*) | 0 | 0.052 | 0.267 | 90.08 |
| BL+RegBN (*vision + touch*) | 0 | 0.052 | 0.231 | 91.54 |

Table 17: Accuracy of classification results on the synthetic dataset in the presence of a confounder. Comparison of the reference to an approach without (W/O) normalization and several normalization techniques. Due to randomness, we reported the mean and std of results over 100 runs.

| Normalization | Norm. params. (#) | | Experiment I | | Experiment II | |
|---|---|---|---|---|---|---|
| Method | MLP | CNN | MLP | CNN | MLP | CN |
| *reference* | - | - | *87.5* | *87.5* | *75.0* | *75.0* |
| W/O normalization | 0 | 0 | 96.2±0.4 | 96.3±0.3 | 86.7±0.6 | 85.4±0.7 |
| BN | 256 | 112 | 96.4±0.4 | 94.9±0.2 | 83.1±0.8 | 85.2±0.3 |
| GN | 256 | 112 | 96.3±0.3 | 95.7±0.3 | 84.6±0.6 | 86.7±0.4 |
| LN | 256 | 112 | 96.1±0.3 | 95.8±0.2 | 85.1±0.5 | 84.7±0.3 |
| IN | 256 | 112 | 96.4±0.4 | 95.9±0.2 | 84.6±0.4 | 83.2±0.9 |
| MDN | 0 | 0 | 91.9±0.7 | 89.7±0.7 | 79.1±0.6 | 82.2±0.7 |
| PMDN | 2048 | 201,728 | 93.4±0.6 | 92.7±0.5 | 80.8±0.7 | 84.3±0.8 |
| RegBN | 0 | 0 | 87.3±0.9 | 88.2±0.8 | 76.2±0.8 | 76.8±0.9 |

this section indicate that specialized normalization techniques, such as those mentioned, are necessary to effectively handle multimodal data. It is worth noting that metadata frequently incorporates confounding variables and noisy data that may be unknown or difficult to measure. In a complicated scenario, the metadata can not only affect the training procedure but also correlates with the predicted label [31]. Therefore, it is crucial for a network to differentiate and handle such values effectively. In the case of employing the MDN and PMDN approaches, it is obligatory to explicitly declare both the metadata and the corresponding labels. However, unlike MDN and PMDN, there is no requirement to specify the metadata and labels when applying RegBN to multimodal models.

# F Ablation study

Here we investigate the influence of RegBN's parameters on the training and test results.

**Comparing adaptive method for estimating $\lambda_+$ with fixed $\lambda_+$ values**: In Section 3.2, we introduced a recursive method for estimating $\lambda_+$ values in each mini-batch to prevent falling into local minima. This section examines the effects of using fixed $\lambda_+$ values versus adaptively-estimated ones. To this end, we selected the synthetic experiment, wherein the amount of confounding effect is known and measurable. Figure 11 illustrates the results obtained for both Experiments I and II. It is evident that when using fixed $\lambda_+$ values, the confounding effect still persists in the results. However, the proposed adaptive technique successfully tracks and removes the confounding effect, leading to a smoother training loss and convergence of the multimodal data toward its reference value. Furthermore, the validation accuracy obtained with the adaptive method demonstrates a narrower range of perturbation in accuracy, indicating the effectiveness of this approach in removing the confounding effect. The adaptive technique proves to be efficient in addressing the challenges posed by confounding factors and achieving more reliable and stable results in the presence of such effects.

**Varaiation of $\lambda_+$ values over mini-batches**: As discussed in the aforementioned experiment and also in Section 3, the projection matrix in RegBN is estimated in every mini-batch (due to the adaptive estimation of $\lambda_+$) and updated recursively. To visualize the evolution of the estimated $\lambda_+$ values throughout the training process, Figure 12 displays a box plot representing the distribution of computed $\lambda_+$ values over epochs in the synthetic dataset. In both Experiment I and Experiment II, the box and whisker plots exhibit a noticeable reduction in length as the multimodal model is trained further. Larger whiskers and wider box plots are observed during the initial epochs, indicating a wider spread of $\lambda_+$ values. However, as the training progresses, the $\lambda_+$ values converge across the epochs, resulting in shorter whiskers and box plots. The $\lambda_+$ values become increasingly consistent and less variable as long as the multimodal model converges.

**Batch size**: In this study, a fixed batch size of 50 was utilized for all experiments with RegBN. Figure 13 presents the influence of batch size on the binary and 5-class accuracy obtained with MulT [55] with/without RegBN in the CMU-MOSI dataset. The figure indicates that batch sizes of 50 and above yield more favorable outcomes when employing RegBN as a normalization method. On the other hand, when using smaller batch sizes, the desired outcomes may not be achieved due to two underlying reasons. Firstly, for RegBN to accurately estimate and optimize its projection matrix, a

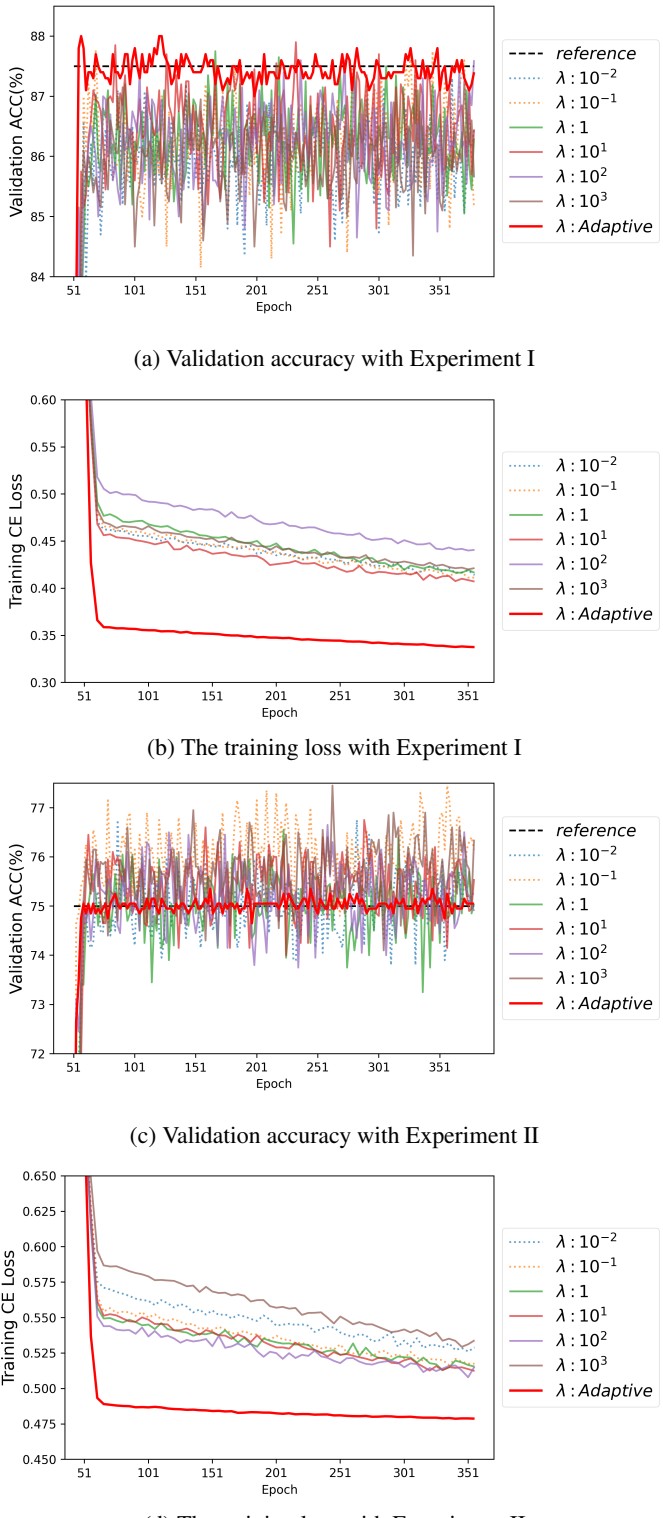

(a) Validation accuracy with Experiment I

(b) The training loss with Experiment I

(c) Validation accuracy with Experiment II

(d) The training loss with Experiment II

Figure 11: Impact of fixed and adaptively-estimated $\lambda_+$ values in RegBN on the synthetic dataset.

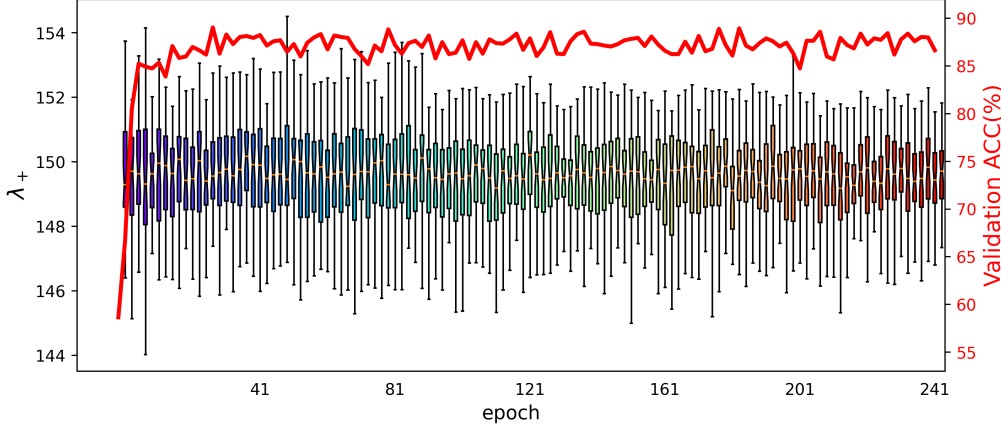

(a) Range of $\lambda_+$ values in Experiment I

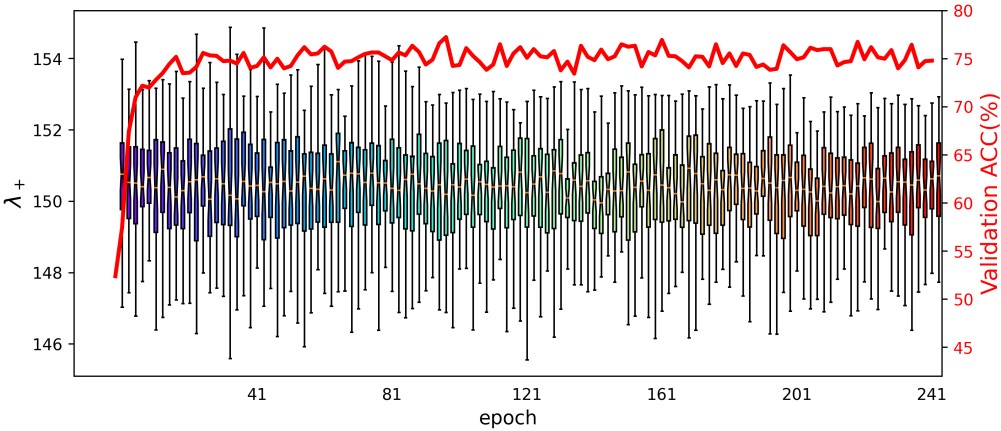

(b) Range of $\lambda_+$ values in Experiment II

Figure 12: Evolution of the distribution and variability of the estimated $\lambda_+$ values across epochs in the synthetic dataset using boxplots. The estimated $\lambda_+$ values exhibit a progressively decreasing range as the training progresses.

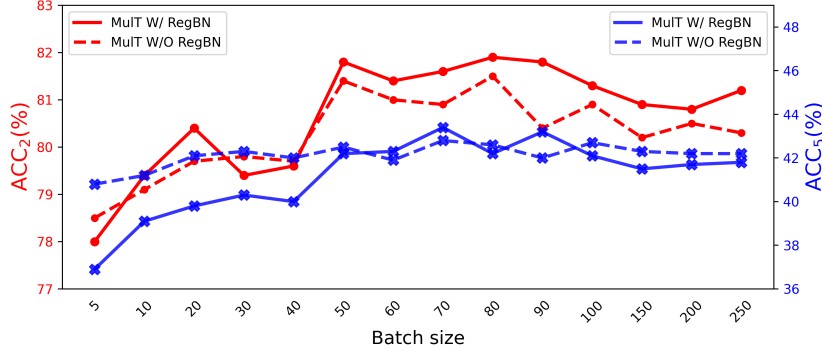

Figure 13: Effect of batch size on the accuracy of MulT [55] with/without RegBN in the CMU-MOSI dataset.

Table 18: Effect of learning rates and maximal steps on the accuracy of MulT [55] with RegBN in the CMU-MOSI dataset.

| Learning rate (#Max. Iterations=25) | $Acc_2\uparrow$ | $Acc_5\uparrow$ | Max. Iterations (#) (Learning rate=1.0) | $Acc_2\uparrow$ | $Acc_5\uparrow$ |
|---|---|---|---|---|---|
| 0.01 | 80.7 | 43.2 | 5 | 80.9 | 41.3 |
| 0.1 | 80.6 | 42.9 | 10 | 81.1 | 42.1 |
| 0.5 | 81.0 | 42.7 | 15 | 81.2 | 41.7 |
| 1.0 | 81.8 | 42.3 | 20 | 81.6 | 42.5 |
| 1.5 | 80.8 | 41.4 | 25 | 81.8 | 42.3 |
| 2.0 | 80.2 | 41.2 | 30 | 81.9 | 42.3 |
| 5.0 | 80.8 | 41.3 | 40 | 81.8 | 42.1 |
| 10.0 | 80.5 | 42.7 | 50 | 81.9 | 42.0 |

sufficient number of observations are required, which is not feasible with small batch sizes. Secondly, there is an interaction between the convergence of the multimodal model and the convergence of RegBN's projection matrix, and they influence each other. The MulT technique does not yield good results for batch sizes of 40 or lower, which directly impacts the performance of RegBN. Generally, to ensure reliable estimation of the projection matrix and subsequent normalization, a batch size of 50 or higher is recommended for RegBN, providing adequate observations. In contrast to MDN, which is susceptible to the negative effects of small batch sizes, RegBN demonstrates effective functionality even with lower batch size values.

**Impact of L-BFGS parameters on RegBN**: In each mini-batch, RegBN seeks the best estimation for $\lambda_+$ via L-BFGS. The learning rate and the maximum number of iterations per optimization step are the important predefined parameters of the L-BFGS algorithm. Table 18 reports the binary accuracy ($Acc_2$) and 5-class accuracy ($Acc_5$) of the MulT method with RegBN on the CMU-MOSI dataset. It is recommended to use a learning rate within the interval of [0.5, 1.0] and configure the maximum number of iterations per step to fall within the range of [20, 30].

# G  Questions and Answers

Q1  How do the concepts of multimodal normalization, consistency, and complementarity interconnect?

*Answer*: **Complementarity** refers to unique information from different modalities, and their combination enriches overall multimodal data interpretation. **Consistency** refers to the degree to which information, patterns, or features align and agree across different modalities' data. In other words, consistency assesses how well the information conveyed by such different modalities corresponds or converges towards a shared understanding. For instance, consistency in the context of vision-text is the textual descriptions accurately represent the content depicted in the images, and vice versa. In the best scenario, information from a modality must reinforce and complement the information from other modalities, leading to a coherent and unified interpretation. **Multimodal nomalisation** like RegBN, which is introduced in this study, aims at making the different modalities' data independent by removing confoundings. As demonstrated by the quantitative and qualitative experimental results, ensuring independence can improve the reliability of analyses and predictions by leveraging the synergies between different types of information while minimizing confounding impacts between modalities.

Q2  Could RegBN improve the *modality imbalance* problem in multimodal databases?

*Answer*: Modality imbalance refers to an uneven distribution of performance or representation among different data modalities like image, audio, etc in multimodal learning. Though the whole multimodal network performance exceeds any single modality, each modality performs significantly below its optimal level. RegBN shows that harnessing modality independence is an efficient means to synergize diverse information types.

Q3  Is RegBN a fusion model?

*Answer*: No, RegBN functions by normalizing input X in relation to input Y, resulting in a normalized X with the same dimensions as the original input X. In fusion, inputs X and Y are combined to generate one or more outputs with distinct content and dimensions. RegBN can be used as a normalization method within the structure of any fusion or neural network.

Q4  Is it feasible to implement RegBN multiple times within a neural network?

*Answer*: Yes, RegBN, like other normalization techniques, can be used multiple times in neural networks. RegBN acts as an independence-promoting layer, so utilizing it multiple times in a row does not substantially alter the feature maps. For a pair of modalities, as outlined in Section A, it is advised to employ RegBN once as a multimodal normalizer within various fusion paradigms. It is important to highlight that in the context of layer fusion (Figure 4c), where RegBN is employed multiple times, the input feature maps at each instance differ from one another. RegBN can be employed as long as its inputs are not mutually independent.