# OpenReview forum: "RegBN: Batch Normalization of Multimodal Data with Regularization"
_NeurIPS.cc/2023/Conference — NeurIPS 2023 poster_

### Official Review · Reviewer_6Kg3 · 2023-07-03

**Soundness:** 2 fair
**Presentation:** 3 good
**Contribution:** 3 good
**Rating:** 5
**Confidence:** 3

**Summary:**

This paper focuses on  mitigate the influence of unwanted variability and bias in multimodal data. To this end, the authors proposed a novel non-parametric regularization technique, i.e., RegBN. The proposed method utilizing Frobenius norm to constrain cross modality consistency, enabling model to discern underlying patterns. Comprehensive experiments show the usefulness of proposed approach in various datasets from different research areas.

**Strengths:**

1. Paper is well-written and follows a good structure.
2. Comprehensive performance evaluation are shown, involving various recent multimodal methods and seven datasets, that proposed approach is working in practice.
3. Mitigating unwanted variability and bias of multimodal data is an important topic and few methods have touched this field. The proposed method is interesting. Improving the reliability of multimodal learning by normalizing multimodal feature is a new perspective.

**Weaknesses:**

Major

- Method: The proposed method mitigates the influence of unwanted variability and bias by minimizes the linear relationship between multimodal layers. In Figure 1, the authors claim that "by leveraging RegBN, these data sources can be rendered independent, thereby enabling a multimodal network to discern underlying patterns and optimize its performance". To the best of my knowledge, exploring consistency and complementarity[1] may be two mainly underlying reason why multimodal model outperforms the unimodal. Does the proposed method ignore to explore the cross-modal consistency?

- Analysis: In line 108, the authors claim that "The goal is to find potential similarities (mainly caused by confounding factors and dependencies during data collection)". In my view, this is an essential assumption for this paper. I think the authors should provide more discussion about why potential confounding factors and dependencies cause these similarities. It could be necessary to give some explanation and citations.
- Comparisons: In line 33 "For instance, the race or gender of speakers in audio classification, backgrounds in video parsing, or the education level of patients in dementia diagnosis is commonly recognized as confounders.". This statements seems very closely to the motivation of DRO[2]. It is appreciated to give some discussion about the relationship between RegBN and DRO.

Minor

- Ambiguous notations: In line 101 and 102, what is the meaning of N and M?

[1] Deep Partial Multi-View Learning, Changqing Zhang, Yajie Cui, Zongbo Han, Joey Tianyi Zhou, Huazhu Fu and Qinghua Hu, IEEE Transactions on Pattern Analysis and Machine Intelligence (IEEE T-PAMI)

[2] Evan Z Liu, Behzad Haghgoo, Annie S Chen, Aditi Raghunathan, Pang Wei Koh, Shiori Sagawa, Percy Liang, and Chelsea Finn. Just train twice: Improving group robustness without training group information. In International Conference on Machine Learning, pages 6781–6792. PMLR, 2021.

**Questions:**

Please address the issues raised in the weakness section.

**Limitations:**

Please address the issues raised in the weakness section.

---

> ### Author Rebuttal · Authors · 2023-08-09
>
> We express our gratitude to the reviewer for the comments and feedback. We have addressed each comment carefully and provided a point-by-point response, as outlined below.
>
> *Question #1: Method: The proposed method mitigates the influence of unwanted variability and bias by minimizes the linear relationship between multimodal layers. In Figure 1,... To the best of my knowledge, exploring consistency and complementarity [1] may be two mainly underlying reason why multimodal model outperforms the unimodal. Does the proposed method ignore to explore the cross-modal consistency?*
>
> **Authors' Response**: In [1], the authors aim to improve performance with missing data. When “views” or data sources are missing for a specific instance, the missing data is imputed. The consistency is used for the generation of the missing information as constraints. In our work, we are not considering missing data, which could be the subject of future work. In the case of complete observations, having the same information multiple times in the network does not provide benefits to the classification and rather makes the network more inefficient (or equivalently called imbalanced modality), as demonstrated in our experiments. RegBN aims at making the different modalities' data independent by removing confoundings. As demonstrated by the quantitative and qualitative experimental results, ensuring independence can improve the reliability of analyses and predictions by leveraging the synergies between different types of information while minimizing confounding impacts between modalities. Thanks to this comment, we have included a concise discussion on consistency, complementarity, and multimodal normalization in the Supplementary’s Q&A section-Q1.
>
> *Comment #2: Analysis: In line 108, the authors claim that "The goal is ..." It could be necessary to give some explanation and citations.*
>
> **Authors' Response**:  We apologize if the text is not clear about the confounders. By definition, a confounder is the common cause of treatment and outcome. Hence, there are similarities between confounder and treatment. We want to remove those similarities (dependencies) to avoid biased results. As an example, the imaging device may impact the characteristics of images, as it is the case for medical images. If the task is to predict diagnosis, and the diagnoses are not balanced across scanners, the model can learn to differentiate scanners instead of disease. Hence, we want to remove the dependence of the scanner on the images.
> We will enhance the clarity of the Introduction Section by incorporating the following sentences:
> “Confounding variables pertain to external factors that introduce bias (either positive or negative)  in the relationship between the variables being studied [1*,2*]. The complexity of confounders emerges from their potential pervasiveness across diverse data modalities. For instance, in image analysis, confounders might encompass lighting variations, while in audio classification, speaker attributes like race or gender can be confounding factors. In video parsing, backgrounds play a role, and in dementia diagnosis, the education level of patients can be a confounder. Furthermore, positive or negative correlations can exist among heterogeneous data that impact the distributions of the learned features [2*, 3*].”
>
> [1*] Elwood JM, editor. Causal Relationships in Medicine. Oxford: Oxford University Press; 1988. p. 332.
>
> [2*] Soleymani et~al.. A survey of multimodal sentiment analysis. Image and Vision Computing, 65, pp.3-14, 2017.
>
> [3*] Manuscript: ref 28
>
> *Comment #3: Comparisons: In line 33 "For instance,...". This statement seems very close to the motivation of DRO[2]. It is appreciated to give some discussion about the relationship between RegBN and DRO.*
>
> **Authors' Response**:  The paper in [2] introduces JTT and compares it to group DRO in the experiments. JTT addresses the problem that classifiers may have an overall high accuracy, but certain groups may have low accuracy. JJT first learns a classifier on the input data and then assigns higher weights to training samples that were misclassified. Subsequently, training a second model with the weighted training set pays more attention to those misclassified samples. This can help with the worst group’s accuracy.
> In RegBN, we address the normalization of multimodal data, which is a very different problem from JTT. We consider multiple input modalities. For instance, consider the example that we would have audio recordings and the gender of the speaker as input. JTT would first train a model and maybe, the samples with the worst accuracy would mainly contain men. By putting more emphasis on the worst samples in the second training, the performance of men's audio recordings may increase. For RegBN, we would have the audio data and gender as inputs. The RegBN layer would now create a latent representation of audio that is independent of the gender variable. In the next step, the classification layer takes normalized audio and gender as its inputs, wherein the audio features are gender-neutral.
>
> *Minor Comment: Ambiguous notations: In lines 101 and 102, what is the meaning of N and M?*
>
> **Authors' Response**:  We thank the reviewer for highlighting this point. In this study,  'f' and 'g’ are N- and M-dimensional feature maps, respectively. We will make the necessary revision to the text to accurately reflect this.
>
> We sincerely appreciate the reviewer's insightful suggestions, and we are committed to integrating them into the final version of the manuscript. We are hopeful that our detailed responses have effectively attended to all of the reviewer's concerns. In the event that any additional questions or points of clarification arise, we kindly invite the reviewer to raise those during the forthcoming reviewer-author discussion period.  Your comment is greatly appreciated, and we look forward to engaging in further discussions during the mentioned timeframe.

---

> > ### Comment · Reviewer_6Kg3 · 2023-08-20
> >
> > Thanks for the authors' response, my main concerns about the motivation and underlying reason of effectiveness has been addressed. Taking other reviewers' comments into account, I tend to increase my score to 5/10.

---

### Official Review · Reviewer_JscA · 2023-07-05

**Soundness:** 3 good
**Presentation:** 3 good
**Contribution:** 2 fair
**Rating:** 5
**Confidence:** 3

**Summary:**

This paper introduces an approach for the normalization of multimodal data called RegBN. The RegBN aligns multimodal features by a learnable projection matrix with regularization in the form of the Frobenius norm. The regularization strength is updated with the L-BFGS optimization algorithm. Extensive experiments on various modalities are conducted to verify the method.

**Strengths:**

- The paper is well-written and easy to follow.
- Aligning multimodal data from the perspective of the normalization method can be promising.
- Extensive experimental results show the effectiveness of the proposed RegBN.

**Weaknesses:**

- The method is not very novel for the reviewer. Essentially, RegBN is an l2 regression with regularization in the form of the Frobenius norm on weight. Moreover, L-BFGS has been widely adopted to update the regularization strength.
- From the extensive experiments, RegBN is actually a multimodal feature fusion module. Other than baselines of normalization methods such as MDN and PMDN, more comparisons with other multimodal feature fusion modules such as [1] and [2] should be provided.
- Why the method is called RegBN. It is not clear how RegBN is connected with the normalization method.

[1] Yikai Wang et al. Learning Deep Multimodal Feature Representation with Asymmetric Multi-layer Fusion.  2021
[2] Yikai Wang et al. Deep Multimodal Fusion by Channel Exchanging. 2020

**Questions:**

Please see the weaknesses. Overall, this paper proposes a simple but effective scheme to better fuse multimodal features. The technical contribution is limited. The experimental results are comprehensive and convincing.  It would be better to provide more comparative experiments as suggested.

**Limitations:**

Yes

---

> ### Author Rebuttal · Authors · 2023-08-09
>
> We appreciate the comments and feedback provided by the reviewer. We would like to address the concerns raised regarding the presentation of our method in the manuscript. We reordered the comments to ensure an efficient and comprehensive response, taking into account overlapping concerns.
>
> *Comments #1&2:
> C1) From the extensive experiments, RegBN is actually a multimodal feature fusion module. Other than baselines of normalization methods such as MDN and PMDN, more comparisons with other multimodal feature fusion modules such as [1] and [2] should be provided.
> C2) Why the method is called RegBN. It is not clear how RegBN is connected with the normalization method.*
>
>
> **Authors' Response**: We have already addressed this concern meticulously earlier in Supplementary-Section A: Multimodal Normalization and Fusion. In that section, we provided a clear discussion about fusion and normalization. In broad terms, fusion and normalization are distinct subjects with unique tasks. RegBN functions by normalizing input X in relation to input Y, resulting in a normalized X with the same dimensions as the original input X. This stands in contrast to fusion, where inputs X and Y are combined to generate one or more outputs with distinct content and dimensions. Hence, RegBN, as a normalization method, can be used in the structure of any fusion or deep learning method. We are well informed about the latest advancements in the fusion domain, and it is noteworthy that the paper [2] recommended by the reviewer had been previously cited as ref [51] in the manuscript. Our manuscript comprehensively covers the nuances of early, layer, and late fusion, as depicted and elaborated in Figure I.1 in Supplementary. We have incorporated a segment of this talk into Supplementary-Section G, addressing Question 3 for more clarification.
>
> *Comment #3: The method is not very novel for the reviewer. Essentially, RegBN is an l2 regression with regularization in the form of the Frobenius norm on weight. Moreover, L-BFGS has been widely adopted to update the regularization strength.*
>
> **Authors' Response**: Based on the previous comment, there may have been confusion by the reviewer about the actual role of RegBN (as a fusion or a normalization layer) and, therefore, also the confusion about its novelty. The other respected reviewers clearly stated the high novelty of RegBN. The novelty of RegBN lies in addressing challenges for multimodal data normalization. Kindly take into account that we do not assert novelty in our use of L-BFGS. We chose it as a practical tool for implementing our method. However, our approach and the objective of this study remain adaptable to any future minimization methodologies.
>
>
> We trust that the provided responses address the reviewer’s concerns regarding RegBN. We have taken diligent measures to ensure comprehensive coverage of RegBN details within both the manuscript and Supplementary. As the reviewer-author discussion period is scheduled from August 10 to August 16, we eagerly await any forthcoming questions and will be delighted to offer further clarification during this time.

---

> > ### Comment · Reviewer_JscA · 2023-08-19
> >
> > Thanks to the authors for the detailed response. The rebuttal addresses my concerns properly. I would like to increase my score to 5.

---

### Official Review · Reviewer_saMu · 2023-07-06

**Soundness:** 3 good
**Presentation:** 3 good
**Contribution:** 3 good
**Rating:** 7
**Confidence:** 4

**Summary:**

The paper introduces a novel approach called RegBN for normalizing multimodal data before fusion in neural networks. The integration of heterogeneous multimodal data poses challenges due to confounding effects and dependencies among different data sources, which can introduce variability and bias. RegBN addresses these issues by using the Frobenius norm as a regularizer term. The method generalizes well across multiple modalities and eliminates the need for learnable parameters, simplifying training and inference. The effectiveness of RegBN is validated on eight databases from various research areas, covering diverse modalities. The proposed method shows broad applicability across different neural network architectures, enabling effective normalization of both low- and high-level features in multimodal models. Overall, RegBN offers a promising solution for improving the performance of multimodal models by addressing normalization challenges.




**Strengths:**

The paper presents a novel idea regarding multi-modal fusion. The proposed method is model-agnostic, i.e., the authors provide details about how to apply RegBN within different frameworks, including late fusion, layer (intermediate) fusion, and early fusion. This method does not require significant modification of the original framework and can be plugged in regardless of the encoder's architecture.

Extensive experiments are conducted, including eight datasets covering applications in multimedia, affective computing, healthcare diagnosis, and robotics. On all datasets, the proposed method outperformed the considered baseline: vanilla model, or model with normalization in other manner such as BatchNorm or PMDN.

This work provides a relatively unique exploration of the topic of multi-modal learning which I believe is worth presenting to the community.

**Weaknesses:**

Authors conducted experiments between with/without RegBN and also compared with other normalization methods.
It will be interesting to see how does RegBN help regardling of the problem of imbalanced modality utilization, as pointed out in recently papers [1, 2, 3]

[1] Wang, Weiyao et al. “What Makes Training Multi-Modal Classification Networks Hard?” 2020 IEEE/CVF Conference on Computer Vision and Pattern Recognition (CVPR) (2019): 12692-12702.
[2] Huang, Yu et al. “What Makes Multimodal Learning Better than Single (Provably).” Neural Information Processing Systems (2021).
[3] Wu, Nan et al. “Characterizing and overcoming the greedy nature of learning in multi-modal deep neural networks.” ICML 2022.

**Questions:**

- Typo in the caption of Figure 3. Subfigures are indexed incorrectly.
- The authors should remember to correct the misstatement in the main paper about SMIL for MM-IMDb.




**Limitations:**

Authors mentioned that the current algorithm works only on single GPU.

---

> ### Author Rebuttal · Authors · 2023-08-09
>
> We extend our gratitude for the reviewer's comments and insightful suggestions. Below, we have provided a point-by-point response to each comment, encompassing all concerns, for your careful consideration.
>
> *Comment: Authors conducted experiments between with/without RegBN and also compared with other normalization methods. It will be interesting to see how does RegBN help regardling of the problem of imbalanced modality utilization, as pointed out in recently papers [1, 2, 3]. ([1] Wang, Weiyao et al. “What Makes Training Multi-Modal Classification Networks Hard?” CVPR-2020. [2] Huang, Yu et al. “What Makes Multimodal Learning Better than Single (Provably).” NeurIPS-2021. [3] Wu, Nan et al. “Characterizing and overcoming the greedy nature of learning in multi-modal deep neural networks.” ICML-2022.)*
>
> **Authors' Response**: We thank the reviewer for the insightful comment. Modality imbalance is a common occurrence in multimodal learning. Fortunately, RegBN can present a valuable solution to address the issues arising from modality imbalance. We conducted an experiment utilizing Prototypical Modality Rebalance (PMR)  [1*], one of the latest developed techniques tailored for mitigating modality imbalance within multimodal learning. Our experiment was conducted on the Colored-and-gray MNIST dataset [2*]. Similar to the MNIST dataset, Colored-and-gray MNIST contains 60,000 training samples and 10,000 test samples. Each sample comprises a gray-scale image (as the first modality) and a monochromatic image (as the second modality). The latter exhibits a strong color correlation with its corresponding digit label. As noted in [1*], the monochromatic images within the training set are robustly color-correlated with their respective digit labels. In contrast, the validation set consists of 10,000 samples, wherein the color correlation of monochromatic images with their labels is weaker. The RegBN in this experiment was applied to raw input images. The  results of the test set are as follows:
>
> PMR:	 [acc: 32.97%,   acc_gray: 19.57%  acc_color: 16.26%]
>
> PMR with RegBN: [acc: 37.04%,   acc_gray: 20.61%  acc_color: 21.44%]
>
> We have also attached the training loss and validation accuracy as Figure 1. As shown in the attached plots, the performance of PMR in both training loss and validation score is boosted in the presence of RegBN. RegBN provides PMR with an enhanced capability to be trained deeper (lower loss) by rendering independency between the two modalities. The optimization mechanism of RegBN, detailed in Sections 3.1&3.2, does not allow the multimodal learning to fall into its local minima. This increased freedom contributes significantly to an improvement in PRM performance. We have condensed a portion of this discussion into Question 2 of the Supplementary's Q&A section (Section G).
>
> [1*] Fan et\~al. PMR: Prototypical Modal Rebalance for Multimodal Learning. CVPR, 2023.
> [2*] Kim et\~al.. Learning not to learn: Training deep neural networks with biased data. CVPR, 2019.
>
> *Questions: 1. Typo in the caption of Figure 3. Subfigures are indexed incorrectly. 2. The authors should remember to correct the misstatement in the main paper about SMIL for MM-IMDb.*
>
> **Authors' Response**:  We highly thank the reviewer for reminding us of the typos. We apologize for the errors and we will correct them in our potential final version.
>
> *Limitation: Authors mentioned that the current algorithm works only on single GPU.*
>
> **Authors' Response**: The initial release of RegBN on GitHub, which will also be referenced in the manuscript's final version, features support for a single GPU. Our ongoing efforts involve the development of a multiple-GPU implementation of L-BFGS, aiming to optimize RegBN for training with large-scale datasets. As progress is made, we will update and release this enhanced version on the GitHub repository.
>
> Once again, we thank the reviewer for every comment. Our responses aim to address all concerns. If more questions or suggestions arise, we invite the reviewer to discuss them during the Aug 10-16 reviewer-author period.

---

> > ### Comment · Reviewer_saMu · 2023-08-20
> >
> > I appreciate the authors in especially extending the experiments and I am very excited to see the results presented. Given my expertise and experience in doing very relevant research on the same topic, I highly recommend this paper be accepted.

---

### Official Review · Reviewer_eB1v · 2023-07-12

**Soundness:** 3 good
**Presentation:** 3 good
**Contribution:** 3 good
**Rating:** 5
**Confidence:** 4

**Summary:**

This paper proposes RegBN for the normalization of multimodal data. RegBN uses the Frobenius norm as a regularizer term to address the side effects of confounders and underlying dependencies among different sources of data. The proposed method generalizes well across multiple modalities and eliminates the need for learnable parameters, simplifying training and inference. Experiments on eight databases from five research areas demonstrate the effectiveness of the proposed method.

**Strengths:**

- This paper proposes a novel RegBN method for the normalization of multimodal data.
- Extensive experiments demonstrate the effectiveness of the proposed method.


**Weaknesses:**

- My main concern is that the proposed method lacks comparison with other competitive normalization methods such layer norm and group norm.
- Using eight databases from five research areas is not necessary. The authors should focus on two or three datasets.
- Synthetic multimodal dataset should be provided to thoroughly evaluate the effectiveness of the proposed method.


**Questions:**

See weaknesses.

**Limitations:**

See weaknesses.

---

> ### Author Rebuttal · Authors · 2023-08-09
>
> We express our gratitude for the reviewer's comments and feedback on our study. We have taken into account all the comments and kindly request the reviewer to consider our responses, which we have addressed in a meticulous manner.
>
> *Comment #1: My main concern is that the proposed method lacks comparison with other competitive normalization methods such layer norm and group norm.*
>
> **Authors' Response**: Thank you for your insights. Since traditional normalization techniques such as Batch normalization (BN), layer normalization (LN), and group normalization (GN) are not explicitly designed for multimodal data and confounder removal, resulting in unsatisfactory results in the experiments that were omitted from the main manuscript. Our manuscript prioritizes the presentation of techniques demonstrating acceptable performance or contextual relevance. To ensure transparency and comprehensiveness, we have already included the results of such techniques in specific experiments, notably the synthetic experiment (Section E.6) and the Healthcare domain experiment (Section E.4). This allows for a holistic understanding of the methodology's performance across varying scenarios. We have updated Supplementary-Sections E.4 and E.6-Tables I.7 and I.9 to include LN results (please find the tables in the attachment). As anticipated, the LN outcomes in both experiments are comparable to those of traditional normalization techniques such as BN and GN.
>
> *Comment #2: Using eight databases from five research areas is not necessary. The authors should focus on two or three datasets.*
>
> **Authors' Response**: Our motivation to assess and validate the proposed method across diverse modalities was driven by the abundance of multimodal data such as video, 2D/3D image, text, tabular, audio, etc. The wide range of experiments presented in this study has provided us with a robust platform to showcase the versatility of RegBN—a facet that has been warmly embraced by our fellow reviewers. We also opted for publicly available and community-accepted benchmark datasets for reproducibility. To ensure readability, we presented a condensed result overview in the manuscript while reserving the detailed results in the Supplementary file for clarity. Our hope is that RegBN gains widespread adoption as a normalization method in multimodal learning, and we are keen to observe the performance of RegBN on multimodal data from other sensors.
>
> *Comment #3: Synthetic multimodal dataset should be provided to thoroughly evaluate the effectiveness of the proposed method.*
>
> **Authors' Response**: Sections 4.5 & E.6 provide the results for synthetic datasets with a single channel. To address the synthetic multimodal dataset, we conducted an experiment utilizing Prototypical Modality Rebalance (PMR)  [1*], one of the latest developed techniques tailored for mitigating modality imbalance within multimodal learning. Our experiment was conducted on the Colored-and-gray MNIST dataset [2*]. Similar to the MNIST dataset, Colored-and-gray MNIST contains 60,000 training samples and 10,000 test samples. Each sample comprises a gray-scale image (as the first modality) and a monochromatic image (as the second modality). The latter exhibits a strong color correlation with its corresponding digit label. As noted in [1*], the monochromatic images within the training set are robustly color-correlated with their respective digit labels. In contrast, the validation set consists of 10,000 samples, wherein the color correlation of monochromatic images with their labels is weaker. The RegBN in this experiment was applied to raw input images. The  results of the test set are as follows:
>
> PMR:	 [acc: 32.97%,   acc_gray: 19.57%  acc_color: 16.26%]
>
> PMR with RegBN: [acc: 37.04%,   acc_gray: 20.61%  acc_color: 21.44%]
>
> We have also attached the training loss and validation accuracy. As shown in the attached plots, the performance of PMR in both training loss and validation score is boosted in the presence of RegBN. RegBN provides PMR with an enhanced capability to be trained deeper (lower loss) by rendering independency between the two modalities. The optimization mechanism of RegBN, detailed in Sections 3.1&3.2, does not allow the multimodal learning to fall into its local minima. This increased freedom contributes significantly to an improvement in PRM performance. We have condensed a portion of this discussion into Question 2 of the Supplementary's Q&A section (Section G).
>
> It is worth noting that the Colored-and-gray MNIST dataset should not be considered a multimodal synthetic dataset. However, it does hold some relevance to Comment \#3, as the input images encompass images with varying channel numbers.
>
> [1*] Fan et\~al. PMR: Prototypical Modal Rebalance for Multimodal Learning. CVPR, 2023.
>
> [2*] Kim et\~al.. Learning not to learn: Training deep neural networks with biased data. CVPR, 2019.
>
> Once more, we extend our gratitude for your valuable comments and feedback on this study. Should any additional questions or clarifications arise, we enthusiastically encourage you to raise them during the upcoming reviewer-author discussion period scheduled from August 10 to August 16.

---

> > ### Comment · Area_Chair_kE7g · 2023-08-20
> >
> > Dear authors,
> >
> > Thank you for providing the response. I acknowledge that this response has been read and will be fully considered.
> >
> > Regards, AC

---

### Official Review · Reviewer_Fqi7 · 2023-07-29

**Soundness:** 4 excellent
**Presentation:** 3 good
**Contribution:** 3 good
**Rating:** 6
**Confidence:** 3

**Summary:**

The paper presents a novel regularization approach tailored for multimodal/heterogenous data fusion, which can be applied off-the-shelf to modern neural network architectures. The approach, namely RegBN, targets the reduction/removal of confounding factors and partial dependencies that are usually present among features extracted from different modalities. To do so, RegBN is implemented as a regularization technique that promotes feature independence across data modalities. The regularization is iteratively optimized over mini-batches using a separate solver not relying on back propagation and gradient descent. This separation is critical as otherwise the network could simply learn to use the confounding factors instead of removing them. Experiments are performed on several multimodal datasets based on different modalities and spanning across different applications, such as multimedia, affective computing, healthcare diagnosis, and robotics. Moreover, a synthetic dataset is designed to further highlight the impact of RegBN. The results showcase a positive impact in all the test cases.

**Strengths:**

1. The approach is novel and grounded with theoretical motivation
2. The method can be plugged off-the-shelf to most multimodal architectures, possibly leading to vaste adoption
3. Wide breadth of experiments that explore several multimodal scenarios
4. Positive results in all scenarios

**Weaknesses:**

1. No integration with recent widely adopted multimodal foundation models, such as CLIP, ImageBind, and many others. Despite I understand that RegBN purpose contrasts with the multimodal alignment objective driving those models, I believe it would be valuable to bring them to the discussion. It would also be beneficial for the positioning of the paper.
2. The boost is sometimes minor, e.g see Table 2, 5, I.3, and in some settings even slightly negative, e.g., see Table 5 and I.4 (video fusion rows). Also, it is unclear the high difference between AV-MNIST results reported in the main paper (~ 99% accuracy) and the ones reported in table I.3 (~71% accuracy). Furthermore, why for the experiments on LLP, training metrics are reported in Table 2 and validation metrics in the supplementary?
3. As RegBN requires a separate optimization, its integration affects the time complexity of the model training. The Authors provides multiple examples on how RegBN could/should be integrated in existing architectures, see Figure I.1, however, in most cases a single RegBN layer is added to the network. It is unclear whether this choice is suboptimal for performance in favor efficiency, or whether it is useless to have multiple RegBN as in I.1c.
4. Minor:
    1. The writing is for sure enjoyable, however there are some parts that could have better explained / could be less ambiguous. For instance, in section 3: _"Ideally, the residual layer ... RegBN minimizes the linear relationship between these layers via: equation (2)"_, but then equation (2) does only contain $f$ and $g$, and thus it is completely tacit the key insight on why solving equation(2) would provide $f_r$ orthogonal to $g$, as well as it is not explicit that $f_r$ corresponds to the normalization of $f$. Moreover, Figure 1 provides a general idea of the impact of RegBN, but it could have been more aligned with the formulation, e.g., by adding $f$, $f_r$, and $g$; and in general the caption could be expanded to better describe it — what does the grey band represent?. Finally the positioning could have been more clear by stating why recent multimodal foundation models like CLIP are out of the scope.
    2. “multimodAl” often mistyped as “multimodEl”; Table I.5, 3rd column, video fusion, (70.6) vs (81.4) seems a bit suspicious; Color gradient is not explained in Figure I.9; wrong x axis label in Figure I.10;

**Questions:**

1. Do the Authors envision any use/application of RegBN for large foundation models, e.g., is it meant for whenever they are employed for downstream tasks?
2. What could be the reason why RegBN affects negatively performance? Is it due to the fact that when evaluating some dataset in-distribution learning spurious correlation might provide a boost?  Please comment also on the last two sentences of weakness # 2.
3. Is it enough to apply RegBN once as shown in I.1a,b,d or might be worth to apply it to several layers?
4. Consider slightly modifying sec. 3 and Figure 1, as well as adding somewhere (introduction or related work) a few positioning sentences wrt foundation models. Fix other typos/mistakes.

**Limitations:**

Societal impact is not discussed. I believe that RegBN could provide a benefit in reducing stereotypical biases that currently affect multimodal models.

---

> ### Author Rebuttal · Authors · 2023-08-09
>
> We express our sincere appreciation to the reviewer for providing valuable comments and suggestions. We have provided a point-by-point response, as outlined below.
>
> *C#1 & Q#1:
> C) No integration with recent widely adopted multimodal foundation models,...
> Q) Do the Authors envision any use/application of RegBN for large foundation models...?*
>
> **Authors' Response**: We highly value the reviewer's insightful suggestion. RegBN is capable of being employed within the architecture of any multimodal technique. In a preliminary experiment involving CLIP, we performed fine-tuning on a pre-trained CLIP model trained on MS-COCO captions [1*]. The results on CIFAR10 are as follows:
> Top-1 Accuracy: CLIP without RegBN: 15.07%, CLIP with RegBN: 16.02%.
> Top-5 Accuracy: CLIP without RegBN: 66.8%, CLIP with RegBN: 67.05%.
> For larger-scale datasets like MS-COCO, layer fusion seems promising to unlock RegBN's full potential (compared to late fusion), and we will prioritize exploring this avenue as you kindly recommended. In accordance with both this comment and the one provided in Question #3, we are motivated to expand the content of the **fourth** paragraph in the Introduction section.
>
> *C#2 and Q#2:
> C) The boost is sometimes minor, ...?*
> Q) What could be the reason why RegBN affects negatively performance?...*
>
> **Authors' Response**: We extend our gratitude for the detailed points. To ensure providing detailed responses, we have categorized and re-ordered the comments into three parts and have addressed each segment as outlined below:
> Part i) AV-MNIST vs. Small AV-MNIST: For enhanced visual representation, we opted to employ small AV-MNIST, consisting of 1,500 raw audio recordings from three speakers and 1,500 MNIST images (manuscript: lines 195-199). Table I.3 presents the outcomes for AV-MNIST. We acknowledge that AV-MNIST is indeed more challenging (compared to small AV-MNIST) due to a preprocessing step that is elaborated in Supplementary-Section C.3. This involves a 75% reduction of visual modality energy via PCA, which results in a noticeable drop in classification performance on the solo visual modality from around 99% on MNIST to 64.3%.
> Part ii) Minor Boost and Fusion Strategy: We recognize the intrinsic challenge in defining and quantifying confounding factors within real-world data as well as in modality imbalance. Different data acquisition settings introduce varying ranges of confounders, which might intensify due to diverse conditions. RegBN's performance is dependent on data specifications. For instance, compression on AV-MNIST exerts a considerable impact on classification results, while a dataset without compression might yield more accurate outcomes (footnote 1). Moreover, the fusion strategy adopted holds a substantial influence on results.
> Part iii) Experiments on LLP: There must have been confusion about the results in Table 2, which are not from the training set but from the test set. We will clarify it in the manuscript. We reported validation results in Supplementary-Section E.2 to showcase RegBN's impact on training from an accuracy perspective.
>
> Footnote 1: In the case of the original AV-MNIST, we reached out to the authors for the dataset, but regrettably, only the compressed version was made available to us.
>
>
> *C#3 & Q#3:
> Comment: As RegBN requires a separate optimization, ... whether it is useless to have multiple RegBN as in I.1c.
> Q:  Is it enough to apply RegBN once ... or might be worth applying it to several layers?*
>
> **Authors' Response**: We are grateful for the insightful comment. From a technical standpoint, RegBN establishes mutual independence between the data of two input modalities. Hence, when applying the RegBN layer successively to the same pair of inputs, the resulting outputs are anticipated to remain independent. Therefore, it is advisable not to utilize RegBN repeatedly over a particular pair of modalities unless there is a shift in the processing context. For instance, modalities X and Y can achieve mutual independence via RegBN, and subsequently, their output can be rendered mutually independent with other modalities as well. As a preliminary experiment, we incorporated RegBN into the MLP network outlined in Section D.6. This integration involved utilizing RegBN K times within the synthetic experiment (Section E.6). For Experiment I, the results for different values of K (K\in{1,2,...,5}) fall within the range of [87.19, 87.43]. Given the inherent randomness of the synthetic dataset, the observed changes in the results (\~0.25%) are relatively minor and opting for `K=1' would be the most suitable choice.
> In the context of layer fusion (Fig. I1.c), where RegBN is employed multiple times, the input feature maps at each instance differ from one another. RegBN can be employed as long as its inputs are not mutually independent. A portion of this part is mentioned as Q4 of the Suppl. Q&A section.
>
> *Minor Cs and Q #3:
> C) The writing is for sure enjoyable, however ...
> Q: Consider slightly modifying sec. 3 and ... Fix other typos/mistakes. *
>
> **Authors' Response**: We acknowledge and apologize for the errors highlighted, all of which will be duly rectified. In line with the reviewer's observations, the accurate value for video fusion in the 3rd column of Table I.5 is indeed 80.6. we have also planned to revise Section 3 regarding the foundation models and modality imbalance. We would like to emphasize our deep appreciation for the reviewer's vigilance in pointing out the inaccuracies and oversights.
>
> *Limitation: Societal impact is not discussed...*
>
> **Authors' Response**: We have planned to enrich the manuscript with a thorough exploration of the societal implications associated with the use of RegBN.
>
> Once again, we deeply appreciate all the comments and suggestions.  We hope we have addressed every comment satisfactorily. We kindly invite the reviewer to raise any additional questions or points during the forthcoming reviewer-author discussion period.

---

> > ### Comment · Reviewer_Fqi7 · 2023-08-19
> >
> > Thanks to the Authors for the additional clarifications. I confirm my score

---

### Author Rebuttal · Authors · 2023-08-09

We sincerely thank the reviewers for their insightful and constructive feedback. We have incorporated their suggestions and responded comprehensively to their comments. Since the reviewer comments do not share common themes, we have addressed them individually. We have incorporated a Q&A section in the Supplementary file that concisely summarizes the discussion related to specific questions raised by the reviewers. This section, provided in one page with 4 questions, is intended to proactively address potential inquiries that readers might have about the same matter.  We hope that our study will be well-received as a valuable contribution to the NeuroIPS' focus on theory & application. We remain available for any further discussions or inquiries the reviewers may have.

It is important to mention that the enclosed file includes figures and tables that have been prepared to address some of the comments from two respected reviewers with the IDs: eB1v and saMu.

Best regards,

Authors

---

### Decision · Program_Chairs · 2023-09-21

**Decision:**

Accept (poster)

**Comment:**

There is a clear consensus among reviewers that the paper introduces a novel approach towards solving the problem of multimodal data normalization that is model-agnostic and reduces mutual information among multimodal features. The paper is well motivated and provides good experimentation and ablative study which adds to the merit of the work. Some questions were raised on the technical aspect of the proposed method and its limitations as well as discussions of the results section e.g. RegBN requiring as separate optimization process along the network training, lack of experimentation on multimodal foundational models.  Authors have addressed several points in their rebuttal which have alleviated most reviewers’ concerns. The AC finds a strong support from reviewers to merit the paper for publication. It is highly encouraged for the authors to take the advantage on the discussions raised by reviewers for their final revision using from both pre-/post-rebuttal phase comments.